# Functional annotation of genetic associations by transcriptome-wide association analysis provides insights into neutrophil development regulation

Yao Yao[1,2], Jia Yang[3], Qian Qin[4], Chao Tang[1], Zhidan Li [1], Li Chen[1], Kailong Li[5], Chunyan Ren[6], Lu Chen[1,8 ✉] & Shuquan Rao [7,8 ✉]

Genome-wide association studies (GWAS) have identified multiple genomic loci linked to blood cell traits, however understanding the biological relevance of these genetic loci has proven to be challenging. Here, we performed a transcriptome-wide association study (TWAS) integrating gene expression and splice junction usage in neutrophils ($N = 196$) with a neutrophil count GWAS ($N = 173,480$ individuals). We identified a total of 174 TWAS-significant genes enriched in target genes of master transcription factors governing neutrophil specification. Knockout of a TWAS candidate at chromosome 5q13.2, *TAF9*, in CD34[+] hematopoietic and progenitor cells (HSPCs) using CRISPR/Cas9 technology showed a significant effect on neutrophil production in vitro. In addition, we identified 89 unique genes significant only for splice junction usage, thus emphasizing the importance of alternative splicing beyond gene expression underlying granulopoiesis. Our results highlight the advantages of TWAS, followed by gene editing, to determine the functions of GWAS loci implicated in hematopoiesis.

---

[1] Key Laboratory of Birth Defects and Related Diseases of Women and Children, Department of Medicine, West China Second Hospital, State Key Laboratory of Biotherapy and Collaborative Innovation Center for Biotherapy, Sichuan University, Chengdu, China. [2] School of Basic Medicine, Chengdu University of Traditional Chinese Medicine, Chengdu, China. [3] Department of Dermatology, University of Californian San Francisco, San Francisco, CA 94110, USA. [4] Molecular Pathology Unit, Center for Cancer Research, Center for Computational and Integrative Biology, Massachusetts General Hospital, Department of Pathology, Harvard Medical School, Boston, MA 02115, USA. [5] Children's Medical Center Research Institute, Department of Pediatrics, Harold C. Simmons Comprehensive Cancer Center, Hamon Center for Regenerative Science and Medicine, The University of Texas Southwestern Medical Center, Dallas, TX 75390, USA. [6] Division of Hematology/Oncology, Boston Children's Hospital, Department of Pediatric Oncology, Dana-Farber Cancer Institute, Harvard Medical School, Boston, MA 02115, USA. [7] Institute of Basic Medical Sciences, Chinese Academy of Medical Sciences & School of Basic Medicine, Peking Union Medical College, Beijing 100005, China. [8] These authors jointly supervised this work: Lu Chen, Shuquan Rao. ✉email: luchen@scu.edu.cn; shuquan.rao@gmail.com

Neutrophils are the most abundant and short-lived human immune cells which form an essential part in innate immune system as an early line of defense against invading microorganisms, and thus are assumed to be key effectors in response to numerous diseases. With about $1 \times 106$ neutrophils produced per second in the bone marrow (BM) of humans, granulopoiesis requires tightly coordinated regulation[1,2]. Neutrophil expansion relies on haemopoietic stem cell (HSC)-derived multipotent progenitors (MPPs) that can give rise to granulocyte-monocyte progenitors (GMPs), further to myeloblast cells (MBs), and finally neutrophils[3,4]. Despite a sophisticated understanding gained primarily from highly penetrant mutations associated with inherited disorders of the hematopoietic system[5], and from model organisms[6,7], many aspects of neutrophil production remain elusive in humans.

At the population level, there is substantial variation in clinically measured blood cell traits, which can manifest as diseases at extreme ends of the spectrum[8]. Identifying genetic variants that drive these differences in blood cell traits in human populations may reveal regulatory mechanisms and genes critical to hematopoiesis. Over the past few years, multiple genome-wide association studies (GWAS) have identified thousands of common and rare variants linked to either neutrophil count (NEUT#) or neutrophil percentage of white blood cells (NEUT%)[9–15]. Nevertheless, identification of causal genes has proven challenging given that most variants are located in non-protein-coding regions and not in linkage disequilibrium (LD) with any non-synonymous coding SNPs[16]. It has been hypothesized that many GWAS-identified associations may function by altering the activity of non-coding biofeatures and thus regulating gene expression[17]. Expression quantitative trait locus (eQTL) analysis can be used to identify associations between risk genotypes and gene expression, and gene expression imputation followed by a transcriptome-wide association study (TWAS) has been extensively proposed as a powerful approach to prioritize candidate risk genes underlying complex traits at known risk regions[18–22]. However, TWAS associations may also be caused by pleiotropy between expression-altering and risk-altering variants or the variants they tag[23,24]. Therefore, experimental validation is needed to establish causality after prioritization of putative target genes by a TWAS.

In this study, we first performed TWAS of NEUT# by integrating gene-expression panel from CD16$^+$ neutrophils[25] and GWAS summary statistics from the UK Biobank and INTERVAL studies ($N = 173,480$)[10]. We prioritized a total of 174 TWAS candidates whose expression is genetically correlated with neutrophil counts, including 56 TWAS candidates, where the GWAS association statistics were below genome-wide significance (lead SNP$_{GWAS}$ $P > 8.31E{-}09$). In addition, we integrated splicing quantitative trait loci (sQTLs) by testing exon junction levels for association with NEUT# (splice junction TWAS, spTWAS), and identified 825 spTWAS associations with NEUT# within 165 unique genes, 89 of which were not captured by gene-level TWAS association. As a proof of concept, we further performed CRISPR/Cas9 gene editing at chromosome 5q13.2 in CD34$^+$ hematopoietic and progenitor cells (HSPCs) for functional assay in order to identify the causal gene. Our study not only identified the causal genes responsible for neutrophil production, but also highlighted the advantages of TWAS, including spTWAS, followed by experimental validation to identify complex trait causal genes.

## Results

### TWAS identifies susceptibility genes associated with neutrophil count (NEUT#).
To identify genes associated with NEUT#, we performed a summary-based TWAS using CD16$^+$ neutrophil

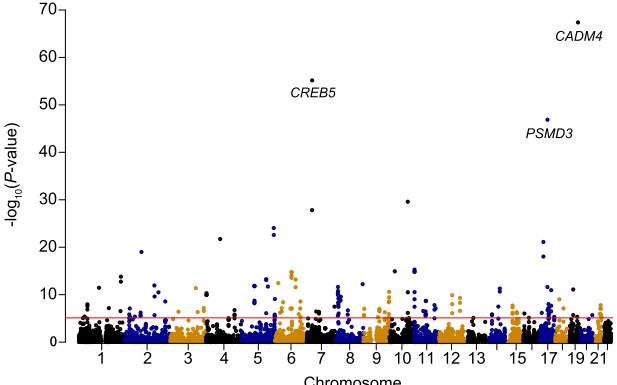

**Fig. 1 Manhattan plot of the transcriptome-wide association study for NEUT# ($N = 173,480$).** A significance threshold of $P = 7.78E{-}06$ was used. NEUT#, neutrophil count.

gene-expression reference panels ($N = 196$)[25] and GWAS summary statistical data from 173,480 participants of European ancestry (the UK Biobank and INTERVAL studies)[10], with the FUSION software[18]. The study design was shown in Supplementary Fig. 1. Briefly, FUSION integrated information from expression reference panels (SNP-expression correlation), GWAS summary statistics (SNP-NEUT# correlation), and linkage disequilibrium (LD) reference panels (SNP-SNP correlation) to assess the association between the cis-genetic component of expression and phenotype (gene expression-NEUT#). In practice, the expression reference panel was used as the LD reference panel, and cis-SNP-expression effect size were estimated with a sparse mixed linear model (e.g., Elastic Net, LASSO and GBLUP) for every gene using local SNPs within 1 Mb flanking window (inclusive).

The TWAS identified a total of 174 transcriptome-wide-significant gene-NEUT# associations surviving Bonferroni correction ($P < 0.05/6430$) (Fig. 1 and Supplementary Data 1). To assess inflation of imputed association statistics under the null hypothesis of no GWAS association, the QTL weights were permuted to empirically determine an association statistic, and the majority of TWAS hits (126/174) were still significant after 1000 permutations ($P < 0.05$) (Supplementary Data 1). Across all TWAS associations, only 26 out of the 174 implicated genes were the nearest gene to the lead SNP within GWAS risk loci. Multiple TWAS hits showed prominent roles in myelopoiesis or neutrophil development, including IKAROS family zinc finger 1 (*IKZF1*, an essential transcription factor expressed throughout hematopoiesis)[26], heterogeneous nuclear ribonucleoprotein K (*HNRNPK*, selectively interacts with the lineage-determining factors, *GATA1* and *SPI1*)[27], nuclear receptor corepressor 1 (*NCOR1*, represses transcription of myeloid-differentiation genes)[28], and so on.

To identify new regulatory regions regulating neutrophil development not identified by GWAS alone, we next examined the overlap between the 174 TWAS associations and genome-wide significant NEUT# GWAS loci[10]. Notably, 56 out of the 174 TWAS hits are located in 34 independent 1 Mb regions, where the GWAS association statistics at TWAS loci were below genome-wide significance (lead SNP$_{GWAS}$ $P > 8.31E{-}09$) (Table 1 and Supplementary Data 1). The discoveries might be either driven by the TWAS aggregating partially independent effects on neutrophil differentiation that operate through a single gene, or due to reduced testing burden for TWAS ($N = 6430$ gene models) compared with that of GWAS (29.5 million imputed variants)[10]. Taken together, these lines of evidence strongly suggested that our TWAS hits may represent causal genes for neutrophil development and emphasized the importance of eQTL from trait related tissue in TWAS.

**Table 1 33 independent transcriptome-wide significant loci at least 500 kb away from any GWAS-identified variants.**

| No. | Region | Jointly significant gene | Type | Best GWAS SNP | Best GWAS P-value | Conditional P-value | TWAS Z-score | TWAS P-value |
|---|---|---|---|---|---|---|---|---|
| 1 | 1p35.2 | LAPTM5 | Protein | rs10737353 | 4.18E−07 | 5.60E−04 | 4.599 | 4.24E−06 |
| 2 | 1q42.12 | CNIH4 | Protein | rs12740255 | 5.77E−08 | 1.000 | −5.122 | 3.03E−07 |
| 3 | 2p23.3 | FAM228A | Protein | rs2543662 | 7.19E−09 | 1.90E−04 | 4.807 | 1.53E−06 |
| 4 | 2p11.2 | ELMOD3 | Protein | rs4832163 | 3.32E−07 | 0.222 | -4.982 | 6.29E−07 |
| 5 | 3p21.31 | SCAP | Protein | rs11920354 | 1.03E−08 | 7.70E−03 | 5.068 | 4.03E−07 |
| 6 | 3q25.1 | MED12L | Protein | rs4603933 | 3.64E−04 | 4.60E−03 | −5.038 | 4.71E−07 |
| 7 | 3q27.1 | ABCC5 | Protein | rs3749440 | 5.13E−08 | 0.202 | −5.391 | 6.99E−08 |
| 8 | 4q31.3 | FAM160A1 | Protein | rs361197 | 7.58E−07 | 1.60E−03 | −5.211 | 1.88E−07 |
| 9 | 5q35.3 | CLK4 | Protein | rs6866344 | 2.06E−07 | 0.130 | −4.794 | 1.63E−06 |
| 10 | 6p21.31 | ZNF76 | Protein | rs914547 | 1.42E−08 | 1.000 | 5.390 | 7.03E−08 |
| 11 | 6q14.1 | TENT5A | Protein | rs915125 | 4.36E−07 | 8.90E−05 | 5.348 | 8.87E−08 |
| 12 | 6q22.32 | CENPW | Protein | rs2184968 | 2.53E−06 | 1.000 | 4.833 | 1.35E−06 |
| 13 | 6q23.3 | PEX7 | Protein | rs9373174 | 2.19E−07 | 8.70E−03 | −4.542 | 5.57E−06 |
| 14 | 7q11.21 | GUSB | Protein | rs937108 | 1.04E−06 | 1.000 | 5.050 | 4.43E−07 |
| 15 | 9p24.2 | VLDLR | Protein | rs1970074 | 1.12E−07 | 4.50E−06 | 4.735 | 2.19E−06 |
| 16 | 9q21.2 | PRUNE2 | Protein | rs680775 | 2.31E−06 | 1.000 | 4.541 | 5.60E−06 |
| 17 | 9q21.32 | HNRNPK | Protein | rs7047907 | 2.90E−08 | 1.000 | 5.316 | 1.06E−07 |
| 18 | 9q32 | SLC31A2 | Protein | rs10121164 | 2.25E−08 | 2.90E−04 | 5.016 | 5.27E−07 |
| 19 | 9q34.3 | FCN1 | Protein | rs11103600 | 4.60E−06 | 0.620 | 4.846 | 1.26E−06 |
| 20 | 11p11.2 | MYBPC3 | Protein | rs10838748 | 5.18E−08 | 4.78E−02 | −5.129 | 2.91E−07 |
| 21 | 11q12.1 | AP001257.1 | lincRNA | rs11230180 | 2.28E−08 | 1.000 | 5.086 | 3.66E−07 |
| 22 | 12q15 | LYZ | Protein | rs2168029 | 1.23E−08 | 0.942 | −6.426 | 1.31E−10 |
| 23 | 14q13.2 | AL121594.1 | Protein | rs12100841 | 2.04E−06 | 1.000 | 4.817 | 1.45E−06 |
| 24 | 14q24.1 | SLC39A9 | Protein | rs12884741 | 2.54E−08 | 1.000 | 5.519 | 3.41E−08 |
| 25 | 15q22.31 | IGDCC4 | Protein | rs4539547 | 6.88E−06 | 0.430 | 4.918 | 8.74E−07 |
| 26 | 16p11.2 | XPO6 | Protein | rs169524 | 6.71E−06 | 0.112 | −4.477 | 7.57E−06 |
| 27 | 17p13.1 | ACAP1 | Protein | rs35776863 | 4.21E−07 | 1.000 | 4.958 | 7.11E−07 |
| 28 | 17q25.1 | SRSF2 | Protein | rs2240772 | 8.66E−07 | 1.50E−02 | 4.567 | 4.94E−06 |
| 29 | 18q21.1 | C18orf25 | Protein | rs8095374 | 1.00E−08 | 2.39E−03 | −5.364 | 8.16E−08 |
| 30 | 19q13.2 | TIMM50 | Protein | rs8107194 | 1.01E−05 | 4.69E−02 | −4.574 | 4.80E−06 |
| 31 | 20q13.33 | ARFRP1 | Protein | rs381331 | 3.04E−07 | n.a. | −4.738 | 2.15E−06 |
| 32 | 21q22.3 | PKNOX1 | Protein | rs11700748 | 1.25E−06 | 0.939 | 5.638 | 1.72E−08 |
| 33 | 21q22.3 | PCNT | Protein | rs2839183 | 1.69E−06 | 2.54E−02 | −4.561 | 5.08E−06 |

**TWAS hits are regulated by neutrophil lineage-specific master transcription regulators (TFs).** Neutrophil lineage commitment is determined by complex interplay of TF regulatory networks, involving PU.1 (encoded by the *SPI1* gene), C/EBPα (CCAAT/enhancer-binding protein α), C/EBPε (CCAAT/enhancer-binding protein ε), runt-related transcription factor 1 (*RUNX1*), Fli-1 proto-oncogene, ETS transcription factor (*FLI1*) and others[2,29]. In order to test whether NEUT# TWAS hit genes were regulated by neutrophil lineage-determining TFs, we referred to our transcriptional regulators (TRs) detection tool, Lisa, which is designed to combine comprehensive human and mouse epigenetic datasets to identify TRs that regulate a query gene set[30]. Several TFs well known to participate in neutrophil specification and expansion were prioritized among a total of 1316 TRs tested, including CEBPA, FLI1, MED12, and RUNX1 (all $P < 7.01E-08$) (Fig. 2a). Relative expression of these TFs across diverse blood cell types were shown in Fig. 2b. To further understand the biologically relevant pathways from the TWAS hits, KEGG pathway and gene ontology analyses were conducted as previously described[31]. Despite of no evidence of pathway enrichment, the NEUT# TWAS hits were significantly enriched for several GO terms, with the most significant being alternative splicing ($P = 2.72E-05$, FDR = 0.04) (Fig. 2c), consistent with our previous finding that alternative splicing plays a crucial role in blood lineage commitment[32].

**Stage-specific expression patterns of TWAS hits during hematopoiesis.** To better understand the potential functions of TWAS hit genes in hematopoiesis, especially in neutrophil lineage commitment, we leveraged BloodSpot (http://servers.binf.ku.dk/bloodspot/) to investigate the expression dynamics of TWAS hits across 7 neutrophil lineage-related cell types and 8 additional cell types[33,34]. A total of 119 out of 174 genes, with available expression data, were grouped into 6 subgroups by hierarchical clustering analysis (Fig. 2d). The majority of TWAS hits (79/119) exhibited high expression in either HSPC (e.g., *U2AF1* and *SOX7*), or common myeloid progenitor (CMP)/GMP (e.g., *PSMD3* and *TIMM50*), or both (e.g., *TAF9* and *ZBTB16*), indicating functional involvement of these genes in early myeloid cell differentiation and/or expansion. Notably, we also found multiple genes that showed exclusively high expression in myeloid or neutrophil stages, such as *IFITM3* (Fig. 2d), which may also account for neutrophil count variation through regulation of neutrophil survival. In addition, multiple genes showed high expression specifically in neutrophil-lineage related and non-related cells types, such as *SEC16A* and *SH3D19* in plasmacytoid dendritic cell, which might be attributed to multi-faceted functions of those genes or shared genetic architecture between neutrophil and other cell types as observed previously[10,35].

**spTWAS prioritize dozens of causal genes independent of TWAS associations.** To identify alternative splicing events essential for neutrophil development, we next performed splice junction TWAS (spTWAS) and identified 825 spTWAS associations with NEUT# in 165 unique genes after Bonferroni correction (Fig. 3a and Supplementary Data 2). More than half of the genes (89/165)

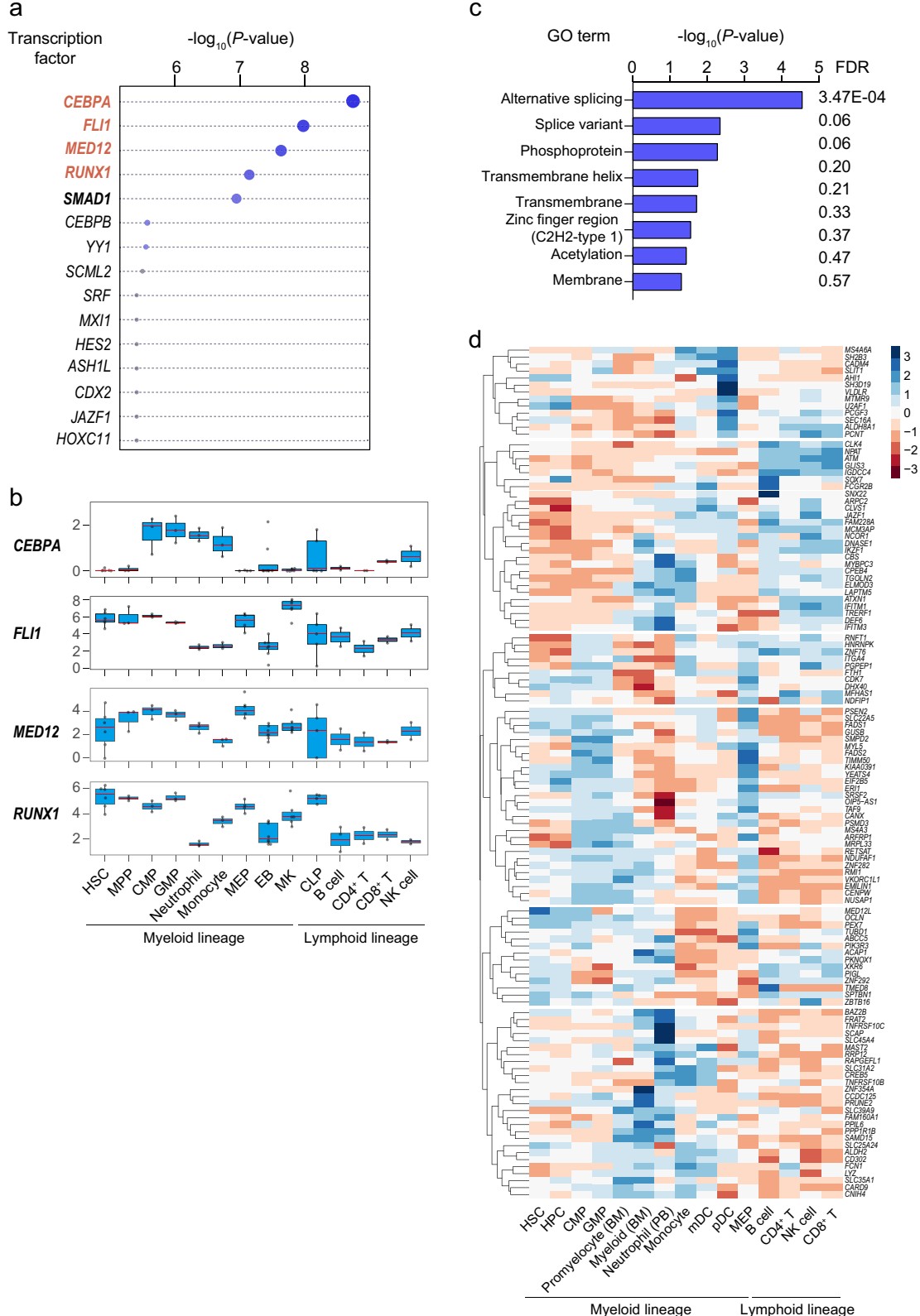

showed no significant gene-level TWAS association, thus representing independent hits (Fig. 3b). Several spTWAS hits are well-known to be involved in myeloid development, including colony stimulating factor 3 receptor (*CSF3R*)[18], NLR family pyrin domain containing 3 (*NLRP3*)[36], Spi-1 proto-oncogene (*SPI1*)[2] and so on. We highlight *CSF3R*, a cytokine that controls the production, differentiation, and function of granulocytes. Multiple spTWAS

associations for the *CSF3R* gene were identified, which fully explained the chr1p34.3 locus (rs3917932 lead $SNP_{GWAS}$ $P = 2.06E$ $-39$) (Fig. 3c). A total of 7 splicing QTLs (spQTL) were shown to regulate CSF3R alternative splicing at different positions, with three of them located within either introns or exons of *CSF3R* (Fig. 3d, e), thus representing potential splicing cis-regulatory variants. Notably, there was no significant eQTLs for overall expression of *CSF3R*

**Fig. 2 Functional assays of TWAS hits partially reveal mechanistic insights. a** Systematic identification of transcriptional regulators (TRs) of TWAS hits (NEUT#) using Lisa (epigenetic Landscape In Silico deletion Analysis and the second descendent of MARGE). TRs with evidence of implication in neutrophil specification and expansion are highlighted in red. **b** Boxplot of TRs gene expression in 14 human hematopoietic cell types. Y-axis represents $\log_2$TPM (Transcripts Per Kilobase Million) value of genes in each sample. Data sources are detailed in "Methods" section. **c** GO enrichment analysis of TWAS hits (NEUT#). For best representation of the results (A and B), $-\log_{10}$(P-values) are reported on a linear scale for best representation of the results. Size of each plot, where available, is correlated with $-\log_{10}$(P-values) (linear model). **d** Heatmap of TWAS hits (NEUT#) in 15 human hematopoietic cell types. Rows, TWAS hits; columns, cell types. Normalized expression data of each gene were downloaded from BloodSpot (http://servers.binf.ku.dk/bloodspot/, HemaExplorer dataset). HSC, hematopoietic stem cell; MPP, multipotential progenitor; CMP, common myeloid progenitor; GMP, granulocyte monocyte progenitor; CLP, common lymphoid progenitor; MEP, megakaryocyte-erythroid progenitor cell; EB, erythroblast; MK, megakaryocyte; HPC, hematopoietic progenitor cell; mDC, myeloid dendritic cell; pDC, plasmacytoid dendritic cell.

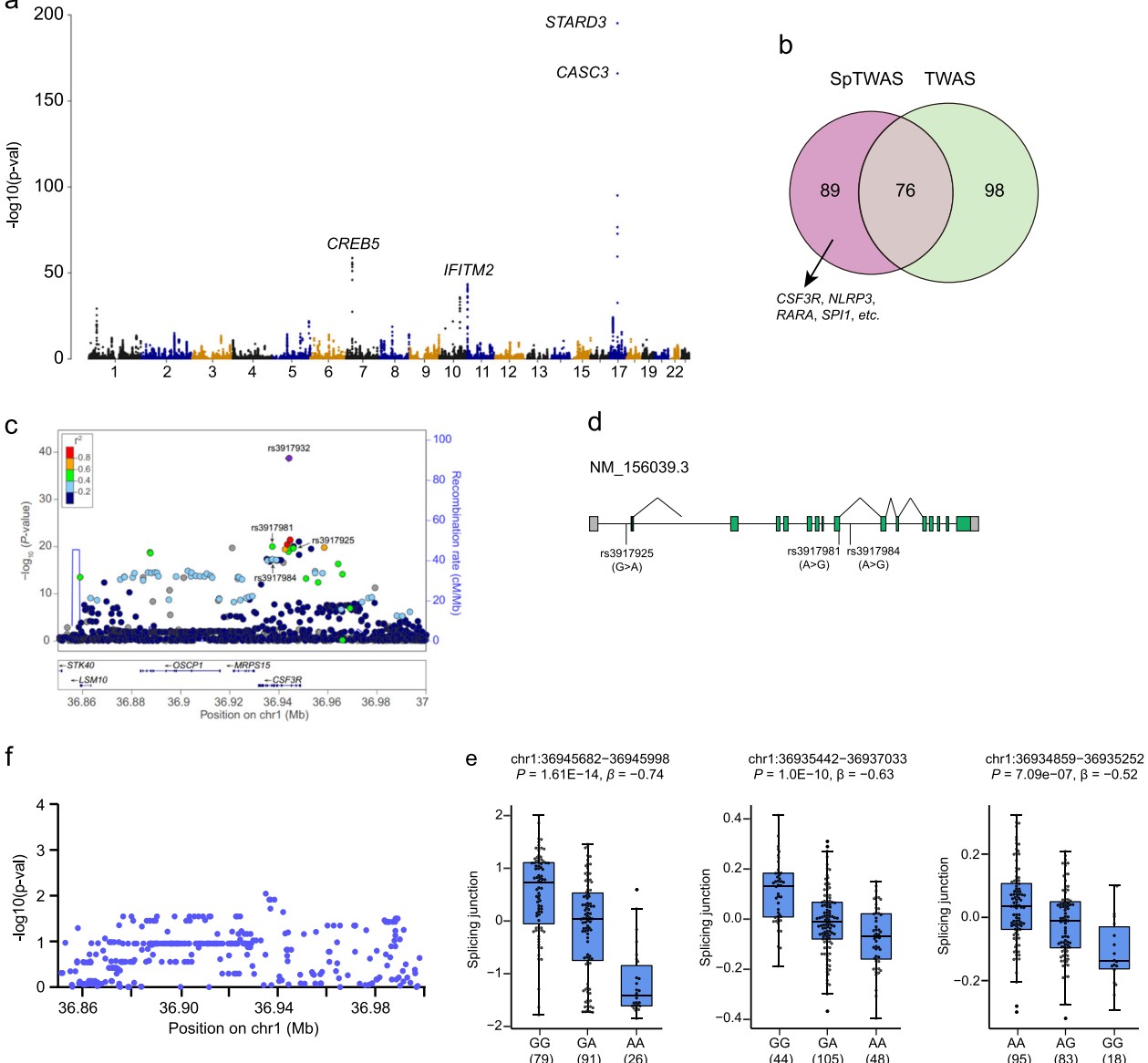

**Fig. 3 Junction spTWAS prioritize dozens of causal genes independent of TWAS associations. a** Manhattan plot of the NEUT# junction spTWAS. P < 1.66Est representation of the result06 (Bonferroni correction, 0.05/30041) was considered significant. **b** Venn diagram of TWAS and junction spTWAS hits for NEUT#. **c** Regional Manhattan Plots for the NEUT# GWAS within 150 kb surrounding the CSF3R gene locus at chromosome 1p34.3. Genotyped and imputed SNPs passing quality control measures are plotted with their P values (as $-\log_{10}$(P-values)) according to their chromosomal positions. Each circle represents one SNP. GWAS-lead SNP rs3917932 and junction spTWAS-lead spQTL were indicated by arrow. The color of each circle indicated the range of pairwise $r^2$ value with rs12187268. The gene positions and transcriptional directions are annotated in the lower panel. **d** Scheme of splice junction and spQTL positions in the CSF3R gene locus. **e** Box plots of normalized junction stratified by representative spQTL genotype (left for rs3917925, and right for rs3917981) in CD16+ neutrophils. **f** Regional cis-eQTL analysis of SNPs on CSF3R at chromosome 1p34.3. Plots show position of SNPs on the x-axis (chromosome length, Mb) and $-\log_{10}$ (P-value) for SNP-gene expression associations on the y-axis shows.

(Fig. 3f), indicating that alternative splicing can independently account for one risk locus. We further performed KEGG pathway and gene ontology analyses, and found that the NEUT# spTWAS hits were significantly enriched in phosphoprotein ($P = 2.70E-08$), alternative splicing ($P = 2.34E-07$) and innate immunity ($P = 1.11E-04$). Overall, these data are consistent with previous suggestions that spQTL explains substantial fraction of complex trait heritability, perhaps even more so than eQTL[32].

**TWAS frequently identify multiple hit genes per locus.** TWAS frequently identify multiple hit genes per locus (Fig. 1), due to correlations among the imputed expression of multiple genes at the same locus, which is comparable to the effect of linkage disequilibrium on SNP-level associations in GWAS[23]. We grouped TWAS hit genes within 1 megabases and found some loci (44/87) with a single hit gene but others with 2–6 candidate genes (Supplementary Fig. 2). To validate whether these signals were due to multiple-associated features or conditionally independent, we performed conditional and joint analyses. The jointly significant genes were summarized in Supplementary Data 1. For example, conditioning on ZNF76 completely explained the variance of loci at chromosome 6 (rs914547 lead $SNP_{GWAS}$ $P = 1.42E-08$, conditioned on ZNF76 lead $SNP_{GWAS}$ $P = 1$) (Supplementary Fig. 3a). It was also found that SMIM8 explained 0.908 of the GWAS signal (rs7774080 lead $SNP_{GWAS}$ $P = 8.29E-17$, conditioned on SMIM8 lead $SNP_{GWAS}$ $P = 1.26E-02$) (Supplementary Fig. 3b). Although conditional and joint analysis can resolve the issue of gene prioritization to some extent, fine-mapping of gene-level association remains challenging, since (1) predicted expression only

imperfectly captures cis expression, owing to both variance and bias in the expression modeling, and (2) the causal gene may fail the test in regions with extensive LD with the eQTLs[18].

**Experimental validation supported TAF9 as the causal gene at chromosome 5q13.2.** Given the possibility of non-causal genes identified by TWAS, experimental validation is essential to establish causality for candidate TWAS associations. The GWAS locus at chromosome 5q13.2 was a notable example, where multiple TWAS hits were identified, and conditional and joint analysis unfaithfully pinpoint the causal gene. The lead SNP rs11745591 at chromosome 5q13.2 showed exclusively significant association with both NEUT# and NEUT% in the same direction (Supplementary Fig. 4). Strong LD was observed for multiple SNPs with the lead SNP rs11745591 (GWAS $P = 5.89E-15$) (Supplementary Fig. 5a). A cluster of 6 TWAS associations were observed responsible for this locus, including TATA-box binding protein associated factor 9 (TAF9, TWAS $P = 1.48E-12$), adenylate kinase 6 (AK6, TWAS $P = 1.44E-12$) and coiled-coil domain containing 125 (CCDC125, TWAS $P = 1.80E-12$), as well as GTF2H2 family member C (GTF2H2C), occludin (OCLN) and cyclin dependent kinase 7 (CDK7) of much less but significant TWAS $P$ values (Supplementary Data 1 and Fig. 4a). Conditioning on AK6 explained 0.787 of the GWAS signal (rs11745591 lead $SNP_{GWAS}$ $P = 5.89E-15$, conditioned on AK6 lead $SNP_{GWAS}$ $P = 3.19E-04$). We next asked whether one TWAS hit gene, despite of non-causality, is prioritized because of correlation with the other causal gene in this region. Interestingly, TAF9 and AK6 showed perfect gene expression, either for total

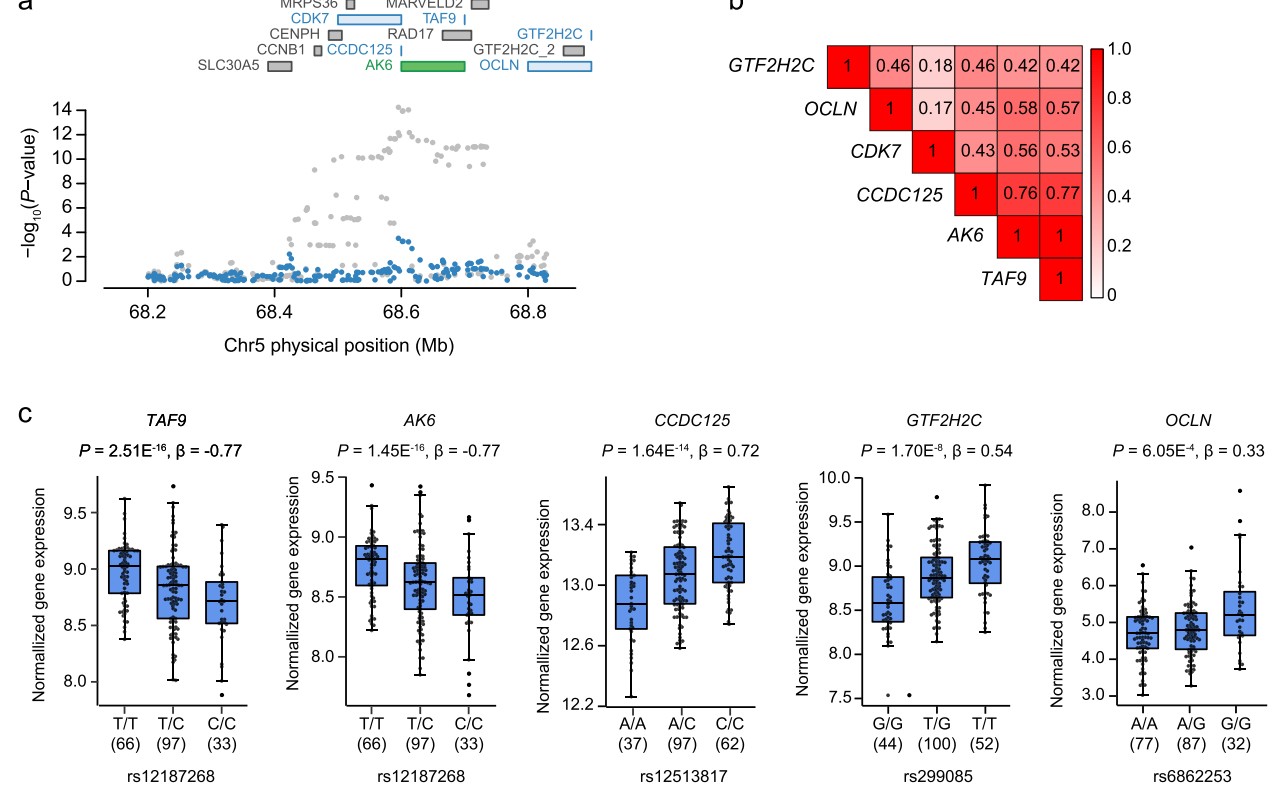

**Fig. 4 TWAS identifies multiple hits at chromosome 5q13.2 for NEUT#. a** Regional association of TWAS hits with NEUT#. The top panel in each plot highlights all genes in the region. The marginally associated TWAS genes are shown in blue and the jointly significant genes are shown in green. The bottom panel shows a regional Manhattan plot of the GWAS associations for each SNP before (gray) and after (blue) conditioning on the predicted expression of the green genes. Two-sided $P$-value computed from GWAS statistics. **b** Predicted gene expression correlation (corr.) matrix among TWAS hits at chromosome 5q13.2. **c** Box plots of normalized gene expression stratified by representative eQTL genotypes in CD16+ neutrophils. eQTLs were identified by using linear regression model and additive genotype effects. NEUT#, neutrophil count; Chr, chromosome.

(corr. = 0.98) (Supplementary Fig. 5b) or predicted gene expression (corr. = 1) (Fig. 4b). Consistent with the expression correlation patterns, highly concordant cis-effects of SNPs on expression were observed between TAF9 and AK6 at chromosome 5q13.2, compared with CCDC125, GTF2H2C, OCLN, or CDK7 (Fig. 4c). Remarkably, multiple SNPs in this region exhibited significant cis-effects on target gene expression, possibly due to strong LD with the causal SNP (Supplementary Fig. 6). Further investigation revealed that TAF9 and AK6 may share the same regulatory regions despite of translation from different reading frames and encoding unrelated proteins (Supplementary Fig. 7), thus explaining the high chance of expression correlation between them.

In order to establish whether TAF9 (or AK6) or any other gene in this locus account for the causality, we performed in vitro functional assays. We designed single guide RNA (sgRNA) targeting each of the six genes and edited human CD34$^+$ HSPCs by electroporation of 3xNLS-SpCas9:sgRNA RNP followed by neutrophil differentiation culture with G-CSF (Fig. 5a). Since CD34$^+$ HSPCs from different donors showed highly variable intrinsic gene expression, we chose CD34$^+$ HSPCs with genotypes predicting relatively high expression of target gene (rs12187268 for TAF9 and AK6, rs12513817 for CCDC125, rs299085 for GTF2H2C, rs6862253 for OCLN, and rs6879078 for CDK7) for functional assay (Fig. 5b, Supplementary Fig. 8), in order to maximize the effect of target gene knockout on neutrophil maturation. All sgRNAs yielded highly efficient gene editing, ranging from 87.0% to 96.3% of indels, respectively (Fig. 5c and Supplementary Fig. 8), which are supposed to disrupt the protein coding sequence or trigger nonsense mediated mRNA decay to knockdown target gene[37]. Since CDK7 is one cell-essential gene, indispensable for human cell fitness[38] and knockdown of CDK7 resulted in much fewer cells compared with controls (Supplementary Fig. 8), we therefore excluded CDK7 for further analysis. All sgRNAs, except for TAF9, editing resulted in significant reduction of mRNA level as compared with neutral locus targeting (Fig. 5d). Given that TAF9 has only one exon and indels-induced frameshifts in coding region can not trigger nonsense mediated decay, we therefore examined the protein level of TAF9 instead. As expected, TAF9 targeting sgRNA resulted in significantly lower protein level as compared with controls (Fig. 5d, Supplementary Fig. 9). TAF9 targeting, but not any other candidate gene targeting, led to significant, albeit mild, increase of neutrophil maturation in vitro (Fig. 5e, f), which was consistent with the GWAS results that rs12187268 C-allele predicted lower TAF9 expression (beta = −0.77, P = 2.51 E−18, Fig. 4c) but more neutrophil counts (effect size = 0.025, P = 3.18E−12, Supplementary Fig. 4).

## Discussion

In the context of hematopoiesis, GWAS studies have identified thousands of variants associated with various blood cell traits[39]. As a result of the rapid progress in this field, the challenge in the post-GWAS era has shifted from identifying genetic regions associated with blood cell traits, to pinpointing the exact regulatory variants and target genes driving each signal, and the cell types in which they act, in order to ultimately understand mechanisms underlying the regulation of hematopoiesis in health and disease[40]. Thorough follow-up efforts at individual loci have identified important regulators of hematopoiesis, such as the key regulator of fetal hemoglobin expression, BCL11A[41,42], however, the low-throughput with which associated genetic variants can be connected to target genes underlying phenotypes continues to pose a problem for gaining biological insights and clinical actionability in hematopoiesis. To address the point, we performed a TWAS of NEUT# using reference expression panel

imputed from CD16$^+$ neutrophils. We identified 174 candidate causal genes, attributable to the previously identified NEUT# GWAS loci and 34 additional risk loci not reported in our previous GWAS[10], thus highlighting the advantages of TWAS in prioritizing candidate causal genes.

Multiple TFs have been identified as essential regulators of myelopoiesis, including PU.1, RUNX1, C/EBPs, interferon regulatory factor-8 (IRF8), lymphoid enhancer binding factor 1 (LEF1), and others[2]. None of these TFs was identified in our TWAS, which is not unexpected considering that (1) dysregulation of key TFs within the myeloid lineage may result in substantial defect of myelopoiesis[43], while (2) common variants identified through GWAS are often of small individual effect. Moreover, gene expression models employed in the current TWAS are based on cis-eQTLs without trans-eQTLs, which may contribute to missing TWAS associations as well. As suggested by Võsa and co-workers, trans-eQTLs often converged on transgenes that are known to play central roles in disease etiology, and might be more informative[44]. Nevertheless, TWAS results are enriched for genes predicted to be regulated by several key TFs according to our non-biased TF prediction model. C/EBPα instructs myeloid differentiation via the priming and activation of myeloid-associated genes in HSPCs[45] and competes for genomic occupancy with other TFs in the myeloid-erythroid progenitor compartment, to favor neutrophilic differentiation[2]. Expression of C/EBPα are determined by a hematopoietic system-specific enhancer element, which is regulated by multiple TFs, including FLI1 and RUNX1[29]. MED12, a regulatory component of the large Mediator complex that enables contact between the general RNA polymerase II transcriptional machinery and enhancer bound regulatory factors[46], and SMAD1[47] were also critical for early myelopoiesis as demonstrated in model organisms. Collectively, these data further emphasized the dominant regulatory effect of these lineage restricted TFs in myelopoiesis and provided clues to investigate the mechanisms of TFs underlying myelopoiesis.

Consistent with the previous study by Gusev et al.[48], our TWAS frequently identified multiple hit genes per locus, with 43 loci containing 2–6 genes per locus, probably due to (1) correlated total expression, (2) correlated predicted expression across individuals between the causal and non-causal genes, (3) sharing of GWAS hits, and (4) multiple causal variants within one GWAS locus[23,35]. The more tightly a pair of genes is co-regulated in cis (the stronger LD between GWAS hits) such as AK6 and TAF9, the more difficult it becomes to distinguish causality on basis of GWAS and expression data. In addition, sequence similarity among distinct genomic regions can lead to errors in alignment of short reads, thus causing false positives in co-expression analyses[49]. Interestingly, shared exonic sequence was observed between TAF9 and AK6 (Supplementary Fig. 7). Although significant correlation of total gene expression between TAF9 and AK6 (corr. = 0.65, P = 2.2E−16) was present after excluding shared exonic sequence, the potential for false positives in expression correlation for other gene pairs should be assessed carefully. Predicted expression correlation tends to be higher than total expression correlation, which may lead to non-causal genes being identified even if the total expression correlation is low.

Using CRISPR/Cas9 gene editing technology and in vitro neutrophil differentiation system, we identified TAF9 as the only TWAS hits driving the GWAS signal at chromosome 5q13.2. Knockdown of TAF9 led to a significant, albeit mild, increase of neutrophil expansion folds. TAF9 encodes one TATA box binding protein (TBP)-associated factor (subunit of transcription factor IID) which acts as a transcriptional coactivator/adapter required at specific promoters. Although no direct evidence had been reported linking TAF9 and neutrophil development, previous studies have revealed how fundamental biological process

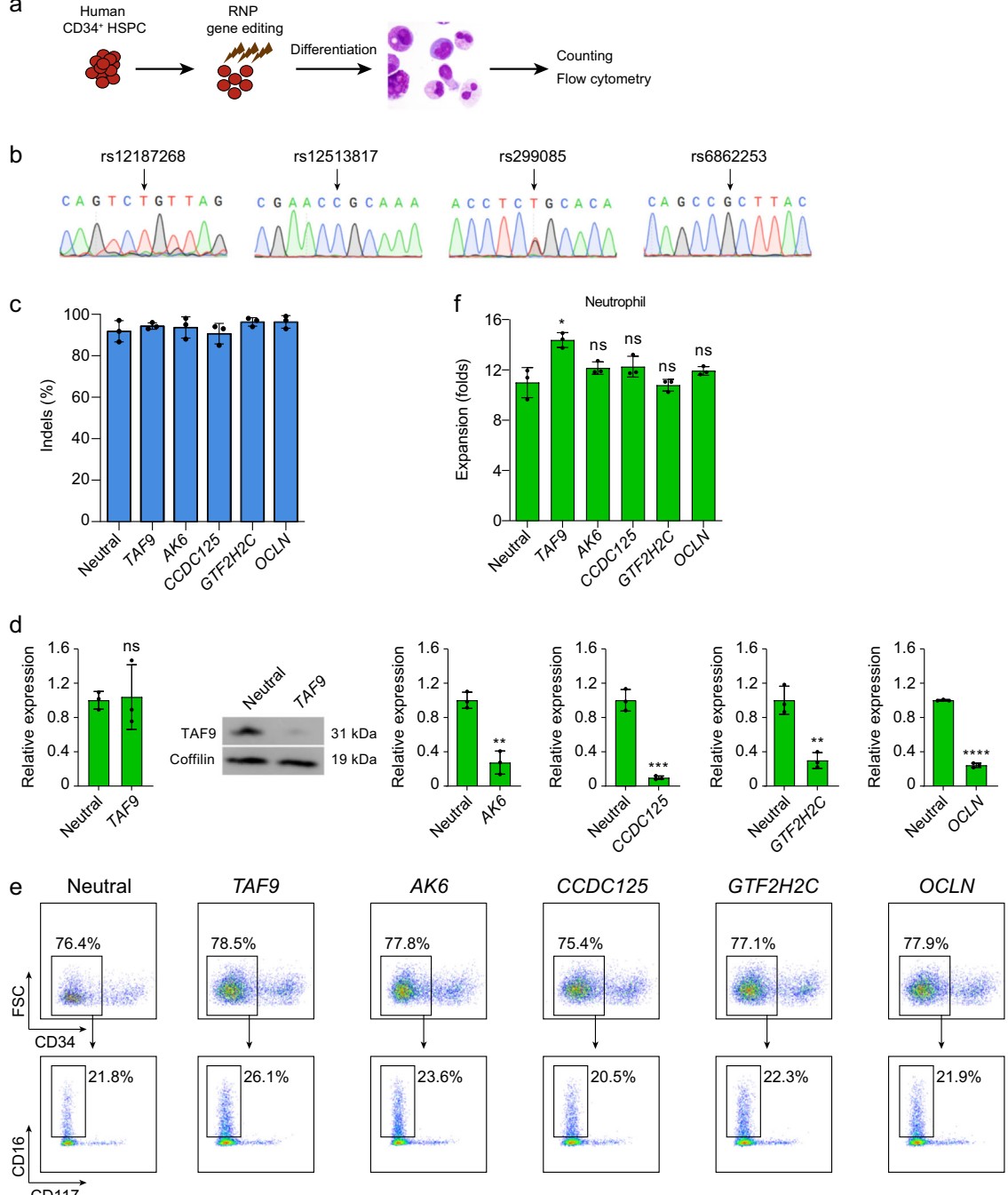

**Fig. 5 Knockdown of *TAF9*, but not other TWAS hits, at chromosome 5q13.2 leads to increase of neutrophil expansion in vitro. a** Schematic of in vitro experimental validation. Human CD34+ HSPCs were obtained from healthy donors and then edited by SpCas9 RNP electroporation (EP) targeting neutral locus and each of TWAS hits at chromosome 5q13.2 (*TAF9, AK6, CCDC125, GTF2H2C, OCLN*, and *CDK*). After 10 days in vitro differentiation into neutrophils, cell cultures were characterized by immunophenotyping. **b** Genomic DNA Sanger sequencing traces show the genotypes of eQTLs (indicated by arrow) corresponding to TWAS hits at chromosome 5q13.2. **c** Editing efficiency in HSPCs following 3xNLS-SpCas9:sgRNA electroporation with indicated sgRNA. Gene edits were measured 6 days after electroporation by Sanger sequencing analysis. $N = 3$ biological replicates for each group. Data was shown as mean ± standard error (SD). **d** Expression level (mRNA or protein) of TWAS hits from RNP edited CD34+ HSPCs 4 days after neutrophil differentiation. mRNA level was determined by RT–qPCR, and protein level was determined by western blot ($N = 3$ biological replicates). $**P < 0.01$; $***P < 0.001$; $****P < 0.0001$; ns, not significant. Data was shown as mean ± SD. **e** Representative flow cytometry of cell cultures (on day 12) from RNP edited CD34+ HSPCs. CD34−CD117−CD16+ cell populations, neutrophils. **f** Expansion of CD34−CD117−CD16+ cell populations 12 days after electroporation calculated by total cell expansion multiplied by fraction determined by flow cytometry ($N = 3$ biological replicates). $*P < 0.05$; ns, not significant. Data was shown as mean ± SD.

can have distinct and unexpected roles in hematopoiesis, like ribosomal proteins in Diamond-Blackfan anemia (DBA)[50]. In addition, despite high expression of TAF9 was observed in CMP/GMP stages, we cannot elucidate the affected cell type using our in vitro neutrophil differentiation system, since progenitor cells only account for a small portion of the whole cell culture and quickly differentiate into the next stage, until mature neutrophils. Further study of TAF9 functions with either TAF9 knockout mice or human CD34⁺ HSPCs xenograft mice models are warranted.

Notwithstanding its significant strengths, our study has some inherent shortcomings due to its design. First, TWAS associations could potentially be confounded by gene expression prediction models which leveraged weighted linear combinations of SNPs. Statistics would be inflated by inclusion of SNPs without regulatory effect of nearby gene expression. Supporting it, difference of gene expression correlation matrix was observed between total and expected expression for the TAF9 locus. Second, the expression reference panel ($N = 6430$ gene models) are based on relatively small sample size. Consequently, there may be features that are important for neutrophil development that we are unable to capture currently using TWAS. Third, some GWAS hits may not affect target gene expression through eQTLs but affect protein structure or alternative splicing instead when located within a gene locus, in which case the target genes will not be captured by TWAS. This could partially explain why there was no candidate causal gene attributable to certain GWAS loci. Notably, TWAS may also fail to identify causal genes when weak eQTLs were attributed to GWAS loci despite of association with the trait. Forth, it should be noted that only individuals of European ancestry were included in the analysis. Again, the CD34⁺ HSPCs used for functional assays are from individuals of European ancestry with specific cis-eQTL genotype (rs12187268 TT predicting higher TAF9 expression compared with TC or CC) in order to maximize the effect of target gene knockdown on neutrophil development. Therefore, whether the results can be generalized to other ethnic groups remains to be investigated. To conclude, TWAS is a powerful statistical method to identify small and large-effect genes responsible for neutrophil differentiation. Further investigation of these genes will provide additional insight into the biology and genetics of neutrophil development.

## Methods

**Summary statistics data of blood cell trait GWAS.** In our previous study, we performed a genome-wide association analysis in the UK Biobank and INTERVAL studies, testing 29.5 million genetic variants for association with 36 red cell, white cell, and platelet properties in 173,480 European-ancestry participants[10]. The traits comprise the main hematological indices of seven cell types reported in a standard clinical full blood count (FBC) analysis (e.g., count of neutrophils as the percentage of white cells, NEUT%) and additional variables derived from them (e.g., neutrophil counts, NEUT#). Phenotype measurement, quality control (QC) and processing of all 36 indices were detailed in our previous study[10]. Due to limitation of trait-related expression data and functional assay platforms, we focused on neutrophil trait in this study.

**Human CD16⁺ neutrophil genotype data.** We previously performed whole-genome sequencing (WGS) (mean read depth, approximately 73) and RNA sequencing (80 million reads per sample) for neutrophils (CD16⁺) from nearly 200 individuals, ascertained to be free of disease and representative of the United Kingdom (UK) population at large[25]. Whole-genome sequencing data (accession number EGAD00001002663) were downloaded from the European Genome-phenome Archive upon request. WGS data was aligned to GRCh37/hg19. Complete details and processing pipeline of WGS data alignment, variant calling, variant QC and filtering were described in our previous study[25], resulting in detection of 7,009,917 variants in total.

**Human CD16⁺ neutrophil RNA-sequencing data processing pipeline and normalization.** CD16⁺ neutrophil RNA-sequencing data (accession number EGAD00001002675) were downloaded from the European Genome-phenome Archive upon request. Details of data generation, processing and quantification, as well as quality control to estimate cross-center and cross-sample identity, can be

found in the original manuscript[25]. Briefly, PCR and sequencing adapters were trimmed using Trimmomatic v0.32 and fastq files after trimming were then aligned to the human reference genome (GRCh37/hg19) using STAR v2.5.3a with the default settings. To quantify and normalize gene expression, we used GenomicAlignments R package to obtain the read counts for each gene annotated in Gencode v19, where the strand information is assigned. The sequencing depth of different samples was then corrected by using library size factor from DESeq2 R package. Sequencing centers and library protocols were used as covariates to adjust batch effects by Combat in sva R package. To reduce the risk of confounding correlation, we applied PEER adjustment on normalized gene expression values after combat batch effect correction using 10 hidden factors. Features showed a unit variance Gaussian distribution after combat and PEER adjustment. Splice junction raw reads were calculated by STAR, of which the output tab-delimited SJ.out.tab table contains unique unmapped and high confidence collapsed splice junctions defining the splice junction start/end as intronic bases. The splice junction expression was obtained after batch-effect correction and PEER with the same way as performed in gene expression.

**Expression quantitative trait loci (eQTL) mapping and splicing QTL (spQTL) mapping of human CD16⁺ neutrophil.** Only subjects that passed genotyping and gene expression QCs were considered for QTL analysis, leaving a sample size of 196. We performed cis-eQTL mapping and cis-spQTL mapping using Matrix EQTL R package. Cis-eQTLs mapping were defined by a cis window of 1 megabase up-stream and down-stream of the gene start site regardless of strand for all PEER-corrected genes.

Cis-spQTL mapping tested association for SNPs within a 1 Mb region surrounding the splice junction boundary using comparable approaches to cis-eQTL analysis.

**Construction and validation of gene/splice prediction models.** Features prediction models, either gene expression or splice junction usage, were created following the TWAS/FUSION method. Matched genotype and PEER-corrected neutrophil gene expression or splice junction reads panel were used to identify a set of variants that influence the features. Here before prediction, variants in neutrophil expression reference panel mentioned above were filtered for quality control using PLINK v1.9 with the options "–maf 1e-10–hwe 1e-6 midp–geno". All remaining SNPs within 1 Mb, i.e., 500 kb distance from each side of the feature boundary, were extracted for estimation of cis-SNP heritability. Features that had nominally significant cis-SNP heritability (likelihood ratio test $P < 0.01$) were retained for model building and TWAS.

We elected to use a heritability-based cutoff rather than specify a cutoff on the cross-validation $R^2$ because the former uses all available data; however, we report both statistics for all associations. Fivefold cross-validation was performed for each feature to estimate the accuracy of prediction model. Gene expression for each fold of the data was hidden in turn, and the full prediction model was then trained on the remaining expression and genetic data. Then the trained model was predicted into the held-out fold samples. This procedure was repeated across all folds to compute the overall cross-validated prediction; an adjusted $R^2$ (and corresponding two-tailed $P$) was then computed between the cross-validated prediction and the measured expression by ordinary least squares. Expression prediction models for protein-coding genes (the MHC region excluded), lncRNAs, microRNAs (miRNAs), processed transcripts, immunoglobulin genes, and T cell receptor genes, according to categories described in the Gencode V19 annotation file were used in TWAS association test. Pseudogenes were not included in further analysis because of potential concerns of inaccurate calling[51]. This resulted in a final total of 6430 gene expression weights (Supplementary Data 3) and 30041 splice junction weights for TWAS association test. Expression weights from the model with the largest $R^2$ were used to compute TWAS association statistics. More details can be found in the original manuscript[18] and other publication[52].

**Transcriptome-wide association study using GWAS summary statistics.** The FUSION software was used to perform the TWAS association tests across all predictive models. Briefly, the association between predicted gene expression and NEUT# was estimated as $Z_{TWAS} = w'Z/(w'Dw)^{1/2}$. To account for multiple hypotheses, TWAS $P$ values from Fusion were Bonferroni-corrected according to the number of genes tested ($N = 6430$) and splice junction tested ($N = 30,041$) in the TWAS association test when assessing statistical significance.

To assess inflation of imputed association statistics under the null of no GWAS association, a permutation test ($N = 1000$ permutations) was conducted to shuffle the QTL weights and empirically determine an association statistic. Permutation was done for each of the significant loci using FUSION. The loci that pass the permutation test demonstrate levels of heterogeneity captured by expression and are less likely to colocalization due to chance. It should be noted that the permutated statistic is very conservative and causal genes could fail to reject the null due to the QTLs having complex and high linkage disequilibrium.

Summary-based conditional analyses for individual SNPs were performed using GCTA conditional and joint analysis[53]. For a given significant TWAS association, the gene expression was predicted into the 1000 Genomes Project European samples to estimate the linkage disequilibrium between the predicted model and

each SNP in the locus. Each GWAS SNP was then conditioned on the predicted model using the linkage disequilibrium estimate to quantify the amount of residual association signal. Stepwise model selection was performed by including each TWAS-associated feature (from most to least significant) into the model until no feature remained conditionally significant.

**TWAS hits expression profiling**. To better understand the potential functions of TWAS hit genes in hematopoiesis, we leveraged BloodSpot (http://servers.binf.ku. dk/bloodspot/) to investigate the expression dynamics of TWAS hits[33,34]. We included 7 cell types along the neutrophil lineage commitment, from HSC, HPC, common myeloid progenitor (CMP) and granulocyte monocyte progenitor (GMP), to promyelocyte, myeloid until neutrophils. As a partial control, 8 additional cell types were also analyzed, including monocyte, myeloid dendritic cell (mDC), plasmacytoid dendritic cell (pDC), megakaryocyte-erythroid progenitor cell (MEP), CD19$^+$ B cell, CD4$^+$ T cell, CD56$^+$ natural killer (NK) cell and CD8$^+$ T cell. Expression data was generated using oligonucleotide microarray chips. The average value of all probes for a given gene was used for clustering and heatmap visualization. More detailed information about gene expression analysis can be found in the original manuscript[33].

TFs expression in 14 human hematopoietic cell types were obtained from the Blueprint Project. In detail, the fastq files were downloaded from EGA upon request, processed and batch corrected as mentioned above. These 14 human hematopoietic cell types are HSC (accession number EGAD00001002316), MPP (accession number EGAD00001002363), common lymphoid progenitor (CLP, accession number EGAD00001002489), common myeloid progenitor (CMP, accession number EGAD00001002478), GMP (accession number EGAD00001002306), megakaryocyte-erythroid progenitor cell (MEP, accession number EGAD00001002433), erythroblast (EB, accession number EGAD00001002358), megakaryocyte (MK, accession number EGAD00001002339), neutrophil (accession number EGAD00001002409), monocyte (accession number EGAD00001002308), B lymphocyte (accession number EGAD00001002438), CD4$^+$ T cell (accession number EGAD00001002348), CD8$^+$ T cell (accession number EGAD00001002295), nature killer cell (NK cell, accession number EGAD00001002321).

*Gene functional enrichment analysis and transcriptional factors (TFs) prediction.* Functional enrichment analysis of TWAS hits was performed by DAVID[31]. To identify the potential transcriptional regulators (TRs) of TWAs hits, we referred to Lisa, (http://lisa.cistrome.org/), which is designed to combine a comprehensive database of human and mouse DNase-seq, H3K27ac ChIP-seq, as well as TR Chip-seq to TRs that regulate a query gene set[30]. The target genes analysis of the top TRs are using rank product of all TR peak and epigenetic regulatory potentials. Top TRs were prioritized based on P-value.

*CRISPR/Cas9 gene editing using RNP electroporation.* The recombinant Streptococcus pyogenes Cas9 with a 6xHistag and c-Myc-like NLS at the N terminus, SV40[40], and nucleoplasmin NLS at the C terminus was expressed and purified as before[54]. The modified CRISPR-Cas9 single guide RNA (sgRNA) was synthesized from IDT. CD34$^+$ HSPCs were thawed and cultured in StemSpan SFEM (STEMCELL Technology) with 1× StemSpan CD34$^+$ expansion supplement (Cat# 02691, STEMCELL Technology) for 24 h. RNP Electroporation was performed using Lonza 4D Nucleofector (V4XP-3032 for 20 µl Nucleocuvette Strips or V4XP-3024 for 100 µl Nucleocuvettes) as the manufacturer's instructions. For 20 µl Nucleocuvette Strips, the RNP complex was prepared by mixing 3× NLS-SpCas9 (200 pmol) and sgRNA (200 pmol, full-length product reporting method) and incubating for 15 min at room temperature immediately before electroporation. HSPCs (0.5–2 ×105 cells) were resuspended in 20 µl P3 solution, mixed with RNP and then transferred to a cuvette for electroporation with program EO-100. The electroporated cells were cultured in StemSpan SFEM with 1× StemSpan CD34$^+$ expansion supplement before induced differentiation. SgRNA spacer sequences were listed in Supplementary Data 4.

**Cell culture and induced neutrophil differentiation of human CD34$^+$ HSPCs**. Human CD34$^+$ HSPCs from mobilized peripheral blood of deidentified healthy donors were purchased from AllCells. CD34$^+$ HSPCs were cultured in StemSpan SFEM medium (StemCell Technologies, Vancouver, Canada) supplemented with 1× StemSpan CD34$^+$ expansion supplement (Cat# 02691, STEMCELL Technology). To induce neutrophil differentiation, the cytokine cocktail of 3 ng/ml G-CSF (unless concentration otherwise indicated), 5 ng/ml IL-3, 100 ng/ml FLT3-Ligand, and 50 ng/ml SCF was supplemented to the culture media for 10 days (+/−1 day) before analysis. GM-CSF of 5 ng/ml was added to the culture media additionally in the first 4 days. All these cytokines were of human origin and purchased from PeproTech (Rocky Hill, NJ).

**Indel frequency analysis by TIDE (Tracking of Indels by DEcomposition)**. Indel frequencies were measured with bulk cell cultures 5 days after RNP electroporation. Briefly, genomic DNA was extracted using the Qiagen Blood and Tissue kit. Genomic region surrounding the sgRNA targeting site was amplified using Hot-StarTaq DNA polymerase (QIAGEN, Cat# 203203) strictly following the

manufactory instructions with variable annealing temperature. PCR products were subject to Sanger sequencing and then TIDE analysis to identifies the major induced mutations within 30 bp away from the projected editing site and accurately determines their frequency in a cell population.

**Cell counting, flow cytometry analysis and apoptosis detection**. Cell counting was performed using either CountBright Absolute Counting Beads, for flow cytometry (ThermoFisher Scientific, Cat# C36950) or hemocytometer. For analysis of surface markers, cells were stained in PBS containing 2% (w/v) BSA, with anti-human CD34 (581), anti-human CD33 (P67.6), and anti-human CD16 (3G8). Flow cytometry data were acquired on a LSRII or LSR Fortessa (BD Biosciences) and analyzed using FlowJo software (Tree Star).

**Determination of gene mRNA and protein level**. Total RNA was extracted from cell cultures 4 days after induced differentiation in vitro (unless indicated elsewhere) using the RNeasy Plus Mini Kit (QIAGEN), and reverse transcribed using the iScript cDNA synthesis kit (Biorad) according to the manufacturer's instructions. Target mRNA was quantified using real-time Q-PCR with GAPDH as an internal control. For quantification of target protein level, whole cell lysates were collected 4 days after induced differentiation in vitro which were then subject to western blot analysis. Anti-TAF9 antibody (Abcam #ab169784, 1:1000 dilution) was used for western blot. All gene expression data represent the mean of at least three biological replicates. All primer sequences were summarized in Supplementary Data 5.

**Statistics and reproducibility**. Statistical analyses were performed using either GraphPad Prism 8 or R package. All pairwise comparisons were assessed using unpaired two-tailed Student's $t$ test unless otherwise indicated in the main text or in the figure legends. Correlation between two groups were assessed by either Pearson correlation or Spearman correlation as indicated in the figure legends. Results were considered significant if the P value was <0.05.

**URLs**. GWAS summary statistics, http://www.bloodcellgenetics.org
EGA website, https://ega-archive.org/
FUSION, http://gusevlab.org/projects/fusion/ GCTA, http://cnsgenomics.com/software/gcta/ GEMMA, http://www.xzlab.org/software.html
PLINK, https://www.cog-genomics.org/plink2/ R, https://www.r-project.org/
Ldsc v1.0.1, https://github.com/bulik/ldsc
MatrixEQTL, https://cran.r-project.org/web/packages/MatrixEQTL/index.html
Lisa, http://lisa.cistrome.org/
Bloodspot database, http://servers.binf.ku.dk/bloodspot/
Trimmomatic software, https://github.com/timflutre/trimmomatic
STAR software, https://github.com/alexdobin/STAR
GenomicAlignments R packge,
https://bioconductor.org/packages/release/bioc/html/GenomicAlignments.html
DESeq2 R pacakage, http://bioconductor.org/packages/release/bioc/html/DESeq2.html
sva R package, https://bioconductor.org/packages/release/bioc/html/sva.html
PEER, https://github.com/PMBio/peer/wiki

**Reporting summary**. Further information on research design is available in the Nature Research Reporting Summary linked to this article.

## Data availability

GWAS summary statistics data of NEUT# and NEUT% were downloaded from http://www.bloodcellgenetics.org. Whole-genome sequencing data of 196 individuals were downloaded from the European Genome-phenome Archive upon request (accession number EGAD00001002663). The RNAseq data in this study are listed in the "Methods" section. TWAS summary statistics are in Supplementary Data 2 and 3. All other materials are available from the corresponding author upon reasonable request.

## Code availability

All the codes are available upon request.

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

## Acknowledgements

We sincerely thank the research team working in the BLUEPRINT project to share the GWAS summary statistics, whole genome sequencing and transcriptome sequencing data. We are grateful to A. Gusev for helpful discussion on splicing TWAS analysis and to S. Coyne for English writing assistance. We also thank those anonymous volunteers for providing CD34+ HSPCs. This work was supported by National Key Research and Development Program of China, Stem Cell and Translational Research [2017YFA0106800 to L.C., 2017YFA0106500 to L.C.]; National Science Fund for Excellent Young Scholars [81722004 to L.C.]; Sichuan Science and Technology Program (No. 2019YFH0137, to Y.Y.) and Chengdu University of Traditional Chinese Medicine (XSGG2019004, to Y.Y.).

## Author contributions

Y.Y., J.Y., and S.R. contributed to the conception and study design. K.L., J.Y., and C.R. designed and performed the experiments. Y.Y., Q.Q., C.T., Li.C., and Z.L. contributed to data collection and analysis. Y.Y., Lu.C. and S.R. contributed to data interpretation and wrote the manuscript.

## Competing interests

The authors declare no competing interests.
