## [Peer Review File · Communications Biology]

Reviewers' comments:

Reviewer #1 (Remarks to the Author):

The authors perform a TWAS integrating gene expression and splice junction coverage for neutrophil count and percentage GWAS. The authors find 56 TWAS-significant genes which were not detected in previous GWAS, and perform downstream enrichment analysis, and validation of a TWAS candidate gene, TAF9 via knockout and assay of neutrophil production in vitro. The authors additionally find 89 genes whose genetically regulated splice junction usage was associated with neutrophil count only via splice-TWAS.

Major comments:

* What is the expected correlation of neutrophil count and neutrophil percentages? What is the genetic correlation? This seems like it would be relevant for the section where neutrophil percentage is described as a partially independent control, but discussion of these two possible correlations was not found. In addition, I did not follow the logic of this statement, "a total of 75 TWAS associations were shared between NEUT# and NEUT%, suggesting that the chance to be false positive signals for those NEUT# TWAS associations is little." How exactly is false positive being defined here?

* I suspect the 0.98 correlation of estimated expression between TAF9 and AK6 may be due to the shared exon, and the use of HTSeq for generating counts (unless the RNA-seq was stranded and HTSeq was provided this information, this wasn't clear). What is the correlation of these two genes when excluding the shared exonic sequence? This may be an important detail to describe with respect to TWAS analysis, and tangentially related to the following publication on false positives in trans eQTL analysis: 10.12688/f1000research.17145.2.

Minor comments:

* Manuscript needs to be proofed for small grammatical errors throughout, e.g. in the Abstract: "Our findings highlight the advantages of TWAS...may facilitate determining the functions..." (missing antecedent of final clause). In Results: "hits have been shown prominent roles...", "despite 62 out of 172 TWAS hits ... were no longer hits...", "To identify extra level beyond...", "...the statistics is highly conservative", "Consistent with previous study...". In Methods, "...splice junction expression were obtained after normalized, batch correct the same way...". This is not a complete list, but some examples.

* "The majority of TWAS hits (126/174) were still significant after 1,000 permutations ($P < 0.05$), suggesting their signal is genuine and not due to chance"

Without multiple correction, 9 of these 126 would have permutation p-value less than 0.05, so I agree the TPR is high among the TWAS hits given the evidence, but it is not expected to be 100% using un-adjusted p-values.

* "As evidence of this model, the TWAS association was stronger than the lead SNP (GWAS association) for 25.5% of TWAS hits that did not overlap a genome-wide-significant SNP, but for only 7.6% of TWAS hits that did overlap a genome-wide-significant SNP"

This comparison seems to be circular as I read it (so the small p-value is not surprising). Paraphrasing: the p-values from TWAS tend to be smaller than the lead SNP p-value when the lead SNP p-value is $> x$ than when the lead SNP p-value is $< x$.

* In the multiple hit genes per locus Results section, the authors discuss conditioning in loci with pairs of genes, but they do not discuss how to approach conditioning with 6 genes for example.

Should all other genes been conditioned upon?

* How exactly were ComBat and PEER combined to correct for batch effects? How many factors were estimated? Details were not sufficient for reproducibility.

* Look at consistency of italicization of gene names when referring to genes and proteins. This did not match our expectation.

* Suggestion to cite ImpG (2014) doi: 10.1093/bioinformatics/btu416 when conducting a summary-based TWAS using expression reference as the LD reference.

Reviewer #2 (Remarks to the Author):

Yao and colleagues performed transcriptome-wide association studies (TWAS) for the neutrophil count and percentage in blood and provided experimental follow-up based on CRISPR/Cas9 to pinpoint the causal gene within a TWAS signal cluster. Specifically, their statistical analysis builds upon their previous works on blood cell traits GWAS and transcriptomic studies of CD16+ neutrophils. They performed TWAS analysis integrating GWAS summary statistics and transcriptomic data to identify genes whose predicted expression or alternative splicing (spTWAS) is associated with neutrophil phenotypes. Via TWAS analysis, they identified additional signals that are below the GWAS significance cutoff. And spTWAS further identified additional signals that are not identified by expression level based TWAS. In the end, they focused on a TWAS locus, 5q13.2, with 6 gene candidates and experimentally confirmed that TAF9 is the potential causal gene by gene editing.

Overall, this paper flows nicely and presents a concrete case study illustrating how TWAS and subsequent gene editing experiments could facilitate causal gene identification of complex traits. But I do have the following comments while reading the manuscript.

1. Colocalization (for instance, COLOC software) of GWAS and eQTL signal is a widely used and accepted alternative of TWAS analysis which may be conservative but it may provide additional insight when TWAS indicates multiple signals within a locus. One would be interested in seeing the colocalization results at least for some representative TWAS locus (for instance 5q13.2).
2. In the first section of "Results", the authors made the point that "the chance to be false positive ... is little". In my opinion, this claim is, first, not so clear, since there wasn't a quantitative statement on how little is little. And secondly, the claim might be too strong since TWAS assumption is usually violated (for instance, the author discussed in "Introduction", pleiotropy exists) which makes the causality hard to establish. So, it might be better to re-frame the corresponding statement in a more precise way.
3. In the second section of "Results", it is unclear to me what message is intended to deliver by the sentence "Consistent with our results, red blood cell-trait-associated GWAS SNPs are also located in DNA sequences recognized by known blood master- transcription-factors (MTF) (e.g., GATA1/TAL1)". It seems unrelated to the mainstream of the text when I read it.
4. In Fig2B and Fig2D, it would be nice if the x-axis could label the cell-type by lineage information and focus more on neutrophil lineage so that it is more readable.
5. In the 4th section of "Results" (spTWAS), I'm curious about the gene set enrichment analysis results just like for expression TWAS. As expression TWAS pointed to splicing, what would these spTWAS genes point to? Would they enrich in the neutrophil development-related pathway? And a

more general question about spTWAS, for those spTWAS genes that were not identified in expression TWAS, do they have expression gene models? By providing more information, one could have more sense on whether the independent hits are real biology or an artifact of statistical power.

6. In Fig3E, why only two of the three exonic/intronic variants are shown (rs3917981 is not shown)? And in Fig3F, it might be informative to overlap with the sQTL p-values for comparison.

7. In the "Discussion", the authors talked about the potential reason why TFs are not identified. Adding upon these, the current TWAS based on predicted expression in cis so that it is expected that these trans-acting signals are missed in general. And efforts have been made to detect trans-signal and one of these approaches test the association between predicted phenotype and observed gene expression (reference: <https://doi.org/10.1101/447367>).

8. Also about "Discussion", I'm confused by "More problematically, many other TWAS hits were also identified, probably due to shared cross-tissue regulatory architecture, which suggests that the strategy of conducting TWAS in a non-trait-related tissue may identify non-causal genes even if there are hits at a locus.". Do "other TWAS hits" mean the signals that are not implicated in neutrophil gene models? If so, how does the "cross-tissue regulatory architecture" play a role? I agree that the TWAS using non-causal/unrelated tissues is an important thing to discuss. So, it would be good to reframe this sentence so that the points get delivered more clearly.

Below are some rather minor comments:

1. In the paragraph about TWAS analysis using non-causal tissue, I found the sentence "..., which implies that statistical power to detect TWAS associations is limited by imputed gene models" confusing. Could the authors clarify in what sense the power is limited by imputed gene models, the quality of the prediction, or the number of the models or something else?

2. Are the expression and splicing models publicly available?

3. Not sure whether there is any difference between sQTL and spQTL. If not, please use one of them.

4. For the last part of the "Results", it would be good to add a transition sentence explaining why 5q13.2 was picked for experimental follow-up.

5. In the last sentence of the "Results", I think there is a typo, FigS8 is meant to be Fig4C instead.

Reviewer #3 (Remarks to the Author):

Yao et al. "Large-scale transcriptome-wide association study identifies new genes and splice variants underlying neutrophil development"

Yao et al., performed TWAS analysis on neutrophil count and replicated some of their results in a TWAS of another neutrophil parameter, neutrophil percentage. They further conducted spTWAS and functionally validated the common gene with their neutrophil count TWAS using CRISPR. This study did extensive statistical analysis, but would be improved by having a more hypothesis

driven approach. As it is, it appears to me like its emphasizing importance of TWAS approach in prioritizing genes involved in neutrophil biology, and their impressive work is more than that. Below are my specific comments

Main comments

1. As they contend in the manuscript in the result section that TWAS signal is much stronger than GWAS signals, I am not sure how only conditioning on lead SNPs will help prioritizes where there are multiple signals in a locus. Other non-lead SNPs might be contributing to the signals that they excluded based on their approach. Moreover, the authors do not mention influence of other variants in models of the specific genes they are prioritizing. The authors beside mentioning the effect of LD on the ability to prioritize a gene, did not clarify LD pattern for the regions they tested vis-à-vis other SNPs that might be in the model used (and if they did, it is not clear from the manuscript). In addition, there are cases where a non-GWAS SNP has larger eQTL effect than the lead SNP. Functional assay done does not preclude the potential for the other genes in the loci from neutrophil biology – because they did not make them controls. The authors did not further explore the reasons (they highlight in the discussion e.g. correlated expressions) for multiple signals in loci.

2. The authors only mention that NEUT% is a derived hematological index....and performed TWAS of NEUT% as a “partially independent control”. I would suggest that they check for level of genetic correlation between the two traits and quantify the level of GWAS signal overlap between the two traits to warrant their claim of independent control.

Minor comments

1. In the abstract, introduction and results the authors state “.....56 of which did not overlap a known GWAS locus” But it does appear that most of these genes that do not overlap with known GWAS loci, do overlap with GWAS loci that are at subgenome-wide threshold based on suppl. Table S1

2. The author state “..several key TFs were predicted to regulate the NEUT# TWAS hits according to our non-biased TF prediction model”. If I understood it right, TWAS results are enriched for genes predicted to be regulated by the TF...The statement and subsequent sentences do not clarify this.

3. I did not completely get the rationale for the TWAS analysis in other tissues by the authors

4. Kindly rewrite the last sentence of the abstract

5. Label some of the signals in Figure 1 and 3....the plain figure without any signal marked does not show any context

6. The authors infer that “.....partially explain why there was no candidate causal gene attributable to certain GWAS loci” But could it also mean that even though the SNP is associated with the trait it might be a weak effect eQTL in aggregate...for the genes in the model

7. I am not sure what evidence the authors use to make the inference that “...that the strategy of conducting TWAS in a non-trait-related tissue may identify non-causal genes even if there are hits at a locus.” Because they have not proved that those hits are non-causal in the manuscript.

8. No page numbers on the manuscript. I am not sure if it is the requirement of the journal but would have made the reviewers commenting easier!

Reviewer #1 (Remarks to the Author):

Major comments:

1. What is the expected correlation of neutrophil count and neutrophil percentages? What is the genetic correlation? This seems like it would be relevant for the section where neutrophil percentage is described as a partially independent control, but discussion of these two possible correlations was not found. In addition, I did not follow the logic of this statement, "a total of 75 TWAS associations were shared between NEUT# and NEUT%, suggesting that the chance to be false positive signals for those NEUT# TWAS associations is little." How exactly is false positive being defined here?

Response:

We thank the reviewer to raise this valuable comment.

NEUT# is a derived hematological index, calculated from white blood cell count and NEUT% with the following formula: $NEUT\# = (NEUT\% \times WBC\#) / 100\%$ where NEUT% and WBC# represent percentage of white cells that are neutrophils and white blood cell count, respectively. The expected correlation of neutrophil count (NEUT#) and neutrophil percentage (NEUT%) depends on the variation of WBC# among populations. However, the exact correlation of neutrophil count and neutrophil percentages was not calculated due to absence of clinical data of blood cell traits.

We estimated the genetic correlation (r_g) between NEUT# and NEUT% using LDSC (LD Score) v1.0.1^{1,2}. The r_g between NEUT# and NEUT% is 0.812 (s.e. = 0.053, $P = 2.3E-24$). In addition, as stated in Astle *et al.*, Cell 2016³, the genetic correlation between NEUT# and NEUT% (r^2) is higher than 0.8.

Given the high genetic correlation between NEUT# and NEUT%, we acknowledged that extra TWAS analysis of the NEUT% (as a partially independent control) would add little evidence to support the causality of NEUT# TWAS hits, we therefore deleted this paragraph in our revised manuscript. In addition, we think that deletion of this section will improve the logic of our manuscript and help to emphasize how TWAS and subsequent gene editing experiments could facilitate causal gene identification of complex traits.

2. I suspect the 0.98 correlation of estimated expression between *TAF9* and *AK6* may be due to the shared exon, and the use of HTSeq for generating counts (unless the RNA-seq was stranded and HTSeq was provided this information, this wasn't clear). What is the correlation of these two genes when excluding the shared exonic sequence? This may be an important detail to describe with respect to TWAS analysis, and tangentially related to the following publication on false positives in trans eQTL analysis: 10.12688/f1000research.17145.2.

Response:

We thank the reviewer to raise this valuable comment.

First, to make it more explicit, we added more details of RNA-sequencing data processing

pipeline and normalization (see Methods section).

Second, as the reviewer suggested, we have excluded the shared exonic sequence and re-analyzed the expression correlation between *TAF9* and *AK6*. Again, we observed significant correlation of total gene expression between *TAF9* and *AK6* ($corr = 0.65$, $P = 2.2E-16$), which was slightly lower than that for total read counts with shared exonic sequence ($corr = 0.98$).

Further investigation revealed that *TAF9* and *AK6* share the same transcriptional regulatory regions despite of translation from different reading frames and encoding unrelated proteins (**Fig. S7**), which may explain the high chance of expression correlation between *TAF9* and *AK6*.

We have added the following discussions in our revised manuscript.

"In addition, sequence similarity among distinct genomic regions can lead to errors in alignment of short reads, thus causing false positives in co-expression analyses⁴. Interestingly, shared exonic sequence was observed between TAF9 and AK6 (Fig. S7). Although significant correlation of total gene expression between TAF9 and AK6 ($corr = 0.65$, $P = 2.2E-16$) was present after excluding shared exonic sequence, the potential for false positives in expression correlation for other gene pairs should be assessed carefully."

Minor comments:

1. Manuscript needs to be proofed for small grammatical errors throughout, e.g. in the Abstract: "Our findings highlight the advantages of TWAS...may facilitate determining the functions..." (missing antecedent of final clause). In Results: "hits have been shown prominent roles...", "despite 62 out of 172 TWAS hits ... were no longer hits...", "To identify extra level beyond...", "...the statistics is highly conservative", "Consistent with previous study...". In Methods, "...splice junction expression were obtained after normalized, batch correct the same way...". This is not a complete list, but some examples.

Response:

According the reviewer's suggestion, we have polished the whole manuscript in terms of grammar.

2. "The majority of TWAS hits (126/174) were still significant after 1,000 permutations ($P < 0.05$), suggesting their signal is genuine and not due to chance"

Without multiple correction, 9 of these 126 would have permutation p-value less than 0.05, so I agree the TPR is high among the TWAS hits given the evidence, but it is not expected to be 100% using un-adjusted p-values.

Response:

We totally agree with the reviewer's comment. To be more precise, we have revised this sentence as below.

"To assess inflation of imputed association statistics under the null of no GWAS association,

the QTL weights were permuted to empirically determine an association statistic, and the majority of TWAS hits (126/174) were still significant after 1,000 permutations ($P < 0.05$) (Table S1)."

3. "As evidence of this model, the TWAS association was stronger than the lead SNP (GWAS association) for 25.5% of TWAS hits that did not overlap a genome-wide-significant SNP, but for only 7.6% of TWAS hits that did overlap a genome-wide-significant SNP"

This comparison seems to be circular as I read it (so the small p-value is not surprising). Paraphrasing: the p-values from TWAS tend to be smaller than the lead SNP p-value when the lead SNP p-value is $> x$ than when the lead SNP p-value is $< x$.

Response:

We thank the reviewer for this valuable comment. In order to avoid confusion, we have deleted this statistical analysis and revise the whole paragraph as below.

"The novel discoveries might be either driven by the TWAS aggregating partially independent effects on neutrophil differentiation that operate through a single gene, or due to reduced testing burden for TWAS ($N = 6430$ gene models) compared with that of GWAS (29.5 million imputed variants)³."

4. In the multiple hit genes per locus Results section, the authors discuss conditioning in loci with pairs of genes, but they do not discuss how to approach conditioning with 6 genes for example. Should all other genes been conditioned upon?

Response:

The purpose to perform conditional and joint analysis is to determine how much variance can be explained by TWAS associations for a given locus, thus providing a fine-mapping strategy of TWAS signals. TWAS loci with multiple hit genes may contain non-causal genes due to correlated expression between non-causal genes and causal genes. Take the *TAF9* TWAS locus for example, despite a total of 6 genes were prioritized as candidate causal genes, only *TAF9* was finally validated to be truly causal gene responsible for this locus. We think conditional analysis upon all the TWAS hits in a given locus may inflate the variance explained by truly causal genes. We thus performed conditional and joint analysis upon the the most significant gene (shown in green in Fig. S4).

5. How exactly were ComBat and PEER combined to correct for batch effects? How many factors were estimated? Details were not sufficient for reproducibility.

Response:

We thank the review for this comment. We have re-organized the description of pipeline used for RNA-sequencing data processing and batch-effect correction. Since we performed similar analyses with those in Chen *et al.* 2016 Cell study, we cited this paper in the present study (in which Chen is one of the corresponding authors) and emphasized any modifications we have

employed.

“Details of data generation, processing and quantification, as well as quality control to estimate cross-center and cross-sample identity, can be found in the original manuscript⁵. Briefly, PCR and sequencing adapters were trimmed using Trimmomatic v0.32 and fastq files after trimming were then aligned to the human reference genome (GRCh37/hg19) using STAR v2.5.3a with the default settings. To quantify and normalize gene expression, we used GenomicAlignments R package to obtain the read counts for each gene annotated in Gencode v19, where the strand information is assigned. The sequencing depth of different samples was then corrected by using library size factor from DESeq2 R package. Sequencing centers and library protocols were used as covariates to adjust batch effects by Combat in sva R package. To reduce the risk of confounding correlation, we applied PEER adjustment on normalized gene expression values after combat batch effect correction using 10 hidden factors. Features showed a unit variance Gaussian distribution after combat and PEER adjustment.”

6. Look at consistency of italicization of gene names when referring to genes and proteins. This did not match our expectation.

Response:

We have revised all the symbol for genes and protein across our manuscript with italicized symbols for genes and non-italicized symbols for proteins.

7. Suggestion to cite ImpG (2014) doi: 10.1093/bioinformatics/btu416 when conducting a summary-based TWAS using expression reference as the LD reference.

Response:

We have cited this paper in our manuscript (see below).

“Expression quantitative trait locus (eQTL) analysis can be used to identify associations between risk genotypes and gene expression, and gene expression imputation followed by a transcriptome-wide association study (TWAS) has been extensively proposed as a powerful approach to prioritize candidate risk genes underlying complex traits at known risk regions 6-10 .”

Reviewer #2 (Remarks to the Author):

1. Colocalization (for instance, COLOC software) of GWAS and eQTL signal is a widely used and accepted alternative of TWAS analysis which may be conservative but it may provide additional insight when TWAS indicates multiple signals within a locus. One would be interested in seeing the colocalization results at least for some representative TWAS locus (for instance 5q13.2).

Response:

We thank the reviewer for this valuable comment. We totally acknowledged that colocalization of GWAS and eQTL signal is an alternative of TWAS analysis (may be more conservative than TWAS). We have also performed colocalization analysis of GWAS and eQTL signals for the chr5q13.2 locus. As we can see below, although the entire GWAS region exhibited strong cis-regulatory effect on *TAF9* expression, inconsistency of lead SNP was observed between GWAS (lead SNP rs11745591) and eQTL analysis (strongest eQTL rs12187268), which might be due to extensive LD among SNPs within this region. In order to avoid confusion of interpreting the results, we therefore didn't put the colocalization analyses in our manuscript.

2. In the first section of "Results", the authors made the point that "the chance to be false positive ... is little". In my opinion, this claim is, first, not so clear, since there wasn't a quantitative statement on how little is little. And secondly, the claim might be too strong since TWAS assumption is usually violated (for instance, the author discussed in "Introduction", pleiotropy exists) which makes the causality hard to establish. So, it might be better to re-frame the corresponding statement in a more precise way.

Response:

We thank the reviewer for this valuable suggestion.

As indicated by Reviewer #1, since extra TWAS analysis of the NEUT%, as a partially independent control (genetic correlation between NEUT# and NEUT% higher than 0.8) would add little evidence to support the causality of NEUT# TWAS hits, we therefore

deleted this paragraph in our revised manuscript. We expected that deletion of this section would further improve the logic of our manuscript and help to emphasize how TWAS and subsequent gene editing experiments could facilitate causal gene identification of complex traits.

3. In the second section of "Results", it is unclear to me what message is intended to deliver by the sentence "Consistent with our results, red blood cell-trait-associated GWAS SNPs are also located in DNA sequences recognized by known blood master- transcription-factors (MTF) (e.g., GATA1/TAL1)". It seems unrelated to the mainstream of the text when I read it.

Response:

We agree with the reviewer on this point, we deleted this sentence in our revised manuscript.

4. In Fig2B and Fig2D, it would be nice if the x-axis could label the cell-type by lineage information and focus more on neutrophil lineage so that it is more readable.

Response:

According to the reviewer's suggestion, we labeled the lineage information in the revised Fig. 2B and 2D.

5. In the 4th section of "Results" (spTWAS), I'm curious about the gene set enrichment analysis results just like for expression TWAS. As expression TWAS pointed to splicing, what would these spTWAS genes point to? Would they enrich in the neutrophil development-related pathway? And a more general question about spTWAS, for those spTWAS genes that were not identified in expression TWAS, do they have expression gene models? By providing more information, one could have more sense on whether the independent hits are real biology or an artifact of statistical power.

Response:

We thank the reviewer for this valuable suggestion. We have performed both KEGG pathway and gene ontology analyses on significant spTWAS genes. Despite of no evidence of pathway enrichment, the NEUT# spTWAS hits were significantly enriched for several GO terms, including phosphoprotein ($P = 2.70E-08$), alternative splicing ($P = 2.34E-07$) and innate immunity ($P = 1.11E-04$). We have added the following contents in the Results section.

"We further performed KEGG pathway and gene ontology analyses, and found that the NEUT# spTWAS were significantly enriched in phosphoprotein ($P = 2.70E-08$), alternative splicing ($P = 2.34E-07$) and innate immunity ($P = 1.11E-04$)."

After careful examination of our gene list, we found that most of spTWAS genes (115 out of 166) have expression gene models. Despite the remaining 51 genes identified by spTWAS didn't have expression gene models, they are probably true considering that many splicing junction events are present for each gene. As we stated in the Methods section, features (both

expression gene models and splicing models) that had nominally significant cis-SNP heritability (likelihood ratio test $P < 0.01$) were retained for model building and TWAS, we believe that some genes that can't pass our strict filtering quality of expression gene model building may still be included in splicing models. We have also included all the TWAS gene models in Supplementary Table 5.

6. In Fig3E, why only two of the three exonic/intronic variants are shown (rs3917981 is not shown)? And in Fig3F, it might be informative to overlap with the sQTL p-values for comparison.

Response:

According to the reviewer's suggestion, we have added splice junctions QTLs for all three variants in the revised Fig. 3E.

As for Fig. 3F, we have tried to compare sQTL and eQTL in one figure. However, since CSF3R has 17 exons, and for each SNP, there are more than one SNP-splicing pair with different P -value, we therefore cannot plot the sQTL P -values according to SNP locations. Below is one example where one SNP corresponds to multiple P -values in our spQTL result.

snps	gene	statistic	pvalue
rs3917932	chr1:36943903-36954255:+	14.9983643	2.87E-34
rs3917932	chr1:36943279-36945033:-	-11.866123	9.01E-25
rs3917932	chr1:36945118-36945587:-	-11.338785	3.42E-23
rs3917932	chr1:36941275-36943205:-	-11.015123	3.14E-22
rs3917932	chr1:36945682-36947078:-	-10.986752	3.82E-22
rs3917932	chr1:36945682-36946100:-	-7.8738724	2.38E-13
rs3917932	chr1:36946280-36947078:-	-7.8102791	3.50E-13
rs3917932	chr1:36945663-36946047:-	-7.8087051	3.53E-13
rs3917932	chr1:36942974-36943473:-	6.60565763	3.71E-10
rs3917932	chr1:36945682-36945998:-	-6.2063452	3.22E-09
rs3917932	chr1:36941275-36945033:-	5.85783488	1.98E-08
rs3917932	chr1:36941189-36943205:-	-5.6863683	4.72E-08
rs3917932	chr1:36945682-36946162:-	-5.5764998	8.15E-08
rs3917932	chr1:36942948-36944088:-	5.4574568	1.46E-07
rs3917932	chr1:36941437-36943205:-	-5.1579911	6.14E-07
rs3917932	chr1:36941569-36943205:-	-5.005586	1.25E-06
rs3917932	chr1:36945682-36946047:-	-4.9480155	1.62E-06
rs3917932	chr1:36946344-36947078:-	-4.85576	2.46E-06
rs3917932	chr1:36933823-36934578:-	-4.6298445	6.69E-06

7. In the "Discussion", the authors talked about the potential reason why TFs are not identified. Adding upon these, the current TWAS based on predicted expression in cis so that it is expected that these trans-acting signals are missed in general. And efforts have been made to detect trans-signal and one of these approaches test the association between predicted phenotype and observed gene expression (reference: <https://doi.org/10.1101/447367>).

Response:

Thanks a lot for raising this valuable suggestion. We have added this in our discussion section.

“Moreover, gene expression models employed in the current TWAS are based on cis-eQTLs without trans-eQTLs, which may contribute to missing TWAS associations as well. As suggested by Vösa and co-workers, trans-eQTLs often converged on trans-genes that are known to play central roles in disease etiology, and might be more informative¹¹.”

8. Also about "Discussion", I'm confused by "More problematically, many other TWAS hits were also identified, probably due to shared cross-tissue regulatory architecture, which suggests that the strategy of conducting TWAS in a non-trait-related tissue may identify non-causal genes even if there are hits at a locus.". Do "other TWAS hits" mean the signals that are not implicated in neutrophil gene models? If so, how does the "cross-tissue regulatory architecture" play a role? I agree that the TWAS using non-causal/unrelated tissues is an important thing to discuss. So, it would be good to reframe this sentence so that the points get delivered more clearly.

Response:

We thank the reviewer for this comment. We have reframed this sentence as below.

“In addition, we found that the majority of neutrophil TWAS hits were dropped out and many non-causal genes were identified (possibly due to different regulatory architectures from those in neutrophils), when using a mechanistically less related tissue with blood, especially neutrophil, such as Testis and Lung. These data strongly emphasize the need for experimental assays to complement TWAS in identifying causal genes at GWAS loci.”

Notably, as suggested by Reviewer #3, we agreed that TWAS analyses in other tissues are not closely related to the topic in the present study. We have deleted this part (*i.e.*, text and supplementary figures) in the revised manuscript.

Below are some rather minor comments:

1. In the paragraph about TWAS analysis using non-causal tissue, I found the sentence "..., which implies that statistical power to detect TWAS associations is limited by imputed gene models" confusing. Could the authors clarify in what sense the power is limited by imputed gene models, the quality of the prediction, or the number of the models or something else?

Response:

The statistical power is limited by the number of imputed gene models. We have revised this sentence as *“which implies that statistical power to detect TWAS associations is limited by the*

number of imputed gene models”.

2. Are the expression and splicing models publicly available?

Response:

The expression and splicing models haven't been uploaded into public database yet. However, we are pleased to share the expression and splicing models upon request.

3. Not sure whether there is any difference between sQTL and spQTL. If not, please use one of them.

Response:

There is no difference between sQTL and spQTL in the present study. Sorry for this confusion. We have changed sQTL to spQTL across our manuscript.

4. For the last part of the "Results", it would be good to add a transition sentence explaining why 5q13.2 was picked for experimental follow-up.

Response:

We thank the reviewer for this suggestion. We have added the following transition sentences.

“Given the possibility of non-causal genes identified by TWAS, experimental validation is essential to establish causality for candidate TWAS associations. The GWAS locus at chromosome 5q13.2 was a notable example, where multiple TWAS hits were identified, and conditional and joint analysis unfaithfully pinpoint the causal gene.”

5. In the last sentence of the "Results", I think there is a typo, FigS8 is meant to be Fig4C instead.

Response:

Thanks very much. We have fixed this in our revised manuscript.

Reviewer #3 (Remarks to the Author):

Yao et al. “Large-scale transcriptome-wide association study identifies new genes and splice variants underlying neutrophil development”

Yao et al., performed TWAS analysis on neutrophil count and replicated some of their results in a TWAS of another neutrophil parameter, neutrophil percentage. They further conducted spTWAS and functionally validated the common gene with their neutrophil count TWAS using CRISPR.

This study did extensive statistical analysis, but would be improved by having a more hypothesis driven approach. As it is, it appears to me like its emphasizing importance of TWAS approach in prioritizing genes involved in neutrophil biology, and their impressive work is more than that.

Below are my specific comments.

Response:

We thank the reviewer for this positive evaluation. In fact, we have recently obtained more data regarding the effect of *TAF9* on myeloid lineage differentiation in xenografted mice with edited CD34⁺ hematopoietic stem and progenitor cells, which provided even more significant phenotype of neutrophil development defects. However, since no *in vivo* experimental data was available about *TAF9* before submission to Communications Biology, we therefore emphasized the TWAS approach, not the biological significance of the *TAF9* locus. In order to more accurately cover all the contents delivered in our manuscript, we intend to change the title as below.

Functional annotation of genetic associations provides insights into neutrophil development regulation

Main comments

1. As they contend in the manuscript in the result section that TWAS signal is much stronger than GWAS signals, I am not sure how only conditioning on lead SNPs will help prioritizes where there are multiple signals in a locus. Other non-lead SNPs might be contributing to the signals that they excluded based on their approach.

Moreover, the authors do not mention influence of other variants in models of the specific genes they are prioritizing. The authors beside mentioning the effect of LD on the ability to prioritize a gene, did not clarify LD pattern for the regions they tested vis-à-vis other SNPs that might be in the model used (and if they did, it is not clear from the manuscript). In addition, there are cases where a non-GWAS SNP has larger eQTL effect than the lead SNP. Functional assay done does not preclude the potential for the other genes in the loci from neutrophil biology – because they did not make them controls. The authors did not further explore the reasons (they highlight in the discussion e.g. correlated expressions) for multiple

signals in loci.

Response:

We thank the reviewer for this valuable suggestion. We tried to answer all these questions one by one.

As they contend in the manuscript in the result section that TWAS signal is much stronger than GWAS signals, I am not sure how only conditioning on lead SNPs will help prioritize where there are multiple signals in a locus. Other non-lead SNPs might be contributing to the signals that they excluded based on their approach.

I totally agree with the reviewer's comment on that both lead and non-lead SNPs may contribute to the signals and thus regulate the causal target gene. As for complex traits, it's becoming increasingly evident that multiple variants can be causal at a single GWAS signal, where they can act in concert in promoters, enhancers or repressors to regulate target gene expression. TWAS aggregates genomic information (both lead and non-lead SNPs) into functionally relevant units that map to genes and their expression. We apologize for confusion in the description of our results section. This gene-based approach combines the effects of many regulatory variants (both lead and non-lead SNPs) into a single testing unit that increases study power and provides more interpretable trait-associated genomic loci^{12,13}. This is why we can discover novel causal genes beyond the original GWAS, especially for those signals that didn't reach genome-wide significance in the original GWAS.

Moreover, the authors do not mention influence of other variants in models of the specific genes they are prioritizing. The authors beside mentioning the effect of LD on the ability to prioritize a gene, did not clarify LD pattern for the regions they tested vis-à-vis other SNPs that might be in the model used (and if they did, it is not clear from the manuscript).

We apologize not to include more details about the TWAS process in our manuscript due to limited space. As we stated before, both lead and non-lead variants have been considered when performing expression gene models, so the predicted expression level of a given gene is integrative effects of multiple variants surrounding the target gene. We have indicated the references where one can obtain more details about the TWAS process in our revised manuscript.

"More details can be found in the original manuscript⁶ and other publication¹⁴."

As shown in Fig. S5 and S6, we can see extensive LD patterns were present at the chromosome 5q13.2.

In addition, there are cases where a non-GWAS SNP has larger eQTL effect than the lead SNP. Functional assay done does not preclude the potential for the other genes in the loci from neutrophil biology – because they did not make them controls. The authors did not further explore the reasons (they highlight in the discussion e.g. correlated expressions) for multiple signals in loci.

We totally agree with the reviewer about this point. Indeed, for most GWAS signals, the strongest eQTLs may not be the lead SNP, and the chromosome 5q13.2 locus we studies is one example. Probably, TWAS and other analyses can't prioritize the truly causal genes responsible in these cases, and this is why we further performed functional experiments, *i.e.*, CRISPR knockout of the potential target genes, to validate their effect on neutrophil differentiation. Our functional experiments were performed with a neutral locus targeting sgRNA as negative control. In order to explore the reasons why multiple hits were prioritized, we have included the following part in the Results section.

“We next asked whether one TWAS hit gene, despite of non-causality, is prioritized because of correlation with the other causal gene in this region. Interestingly, TAF9 and AK6 showed perfect gene expression, either for total (corr. = 0.98) (Fig. S6B) or predicted gene expression (corr. = 1) (Fig. 4B). Consistent with the expression correlation patterns, highly concordant cis-effects of SNPs on expression were observed between TAF9 and AK6 at chromosome 5q13.2, compared with CCDC125, GTF2H2C, OCLN or CDK7 (Fig. 4C). Remarkably, multiple SNPs in this region exhibited significant cis-effects on target gene expression, possibly due to strong LD with the causal SNP (Fig. S6A). Further investigation revealed that TAF9 and AK6 may share the same regulatory regions despite of translation from different reading frames and encoding unrelated proteins (Fig. S7), thus explaining the high chance of expression correlation between them.”

2. The authors only mention that NEUT% is a derived hematological index....and performed TWAS of NEUT% as a “partially independent control”. I would suggest that they check for level of genetic correlation between the two traits and quantify the level of GWAS signal overlap between the two traits to warrant their claim of independent control.

Response:

We thank the reviewer for this valuable suggestion.

As indicated by Reviewer #1 and #2, since extra TWAS analysis of the NRUT%, as a partially independent control (genetic correlation between NEUT# and NEUT% higher than 0.8) would add little evidence to support the causality of NEUT# TWAS hits, we therefore deleted this paragraph in our revised manuscript. We expected that deletion of this section would further improve the logic of our manuscript and help to emphasize how TWAS and subsequent gene editing experiments could facilitate causal gene identification of complex traits.

Minor comments

1. In the abstract, introduction and results the authors state “.....56 of which did not overlap a known GWAS locus” But it does appear that most of these genes that do not overlap with known GWAS loci, do overlap with GWAS loci that are at subgenome-wide threshold based

on suppl. Table S1.

Response:

We thank the reviewer for this valuable comment. We have revised this as below to be more concise.

“Notably, 56 out of the 174 TWAS hits are located in 34 independent 1 Mb regions, where the GWAS association statistics at novel TWAS loci were below genome-wide significance (lead $SNP_{GWAS} P > 8.31E-09$).”

2. The author state “..several key TFs were predicted to regulate the NEUT# TWAS hits according to our non-biased TF prediction model”. If I understood it right, TWAS results are enriched for genes predicted to be regulated by the TF...The statement and subsequent sentences do not clarify this.

Response:

According to the reviewer’s suggestion, we have revised this sentence.

“Nevertheless, TWAS results are enriched for genes predicted to be regulated by several key TFs according to our non-biased TF prediction model.”

3. I did not completely get the rationale for the TWAS analysis in other tissues by the authors

Response:

We agree with the reviewer that TWAS analyses in other tissues are not closely related to the topic in the present study. We have deleted this part (*i.e.*, text and supplementary figures) in the revised manuscript.

4. Kindly rewrite the last sentence of the abstract.

Response:

We have rewritten the last sentence of the abstract.

“Our findings highlight the advantages of TWAS, followed by gene editing in $CD34^+$ HSPCs, to determine the functions of GWAS loci implicated in hematopoiesis.”

5. Label some of the signals in Figure 1 and 3....the plain figure without any signal marked does not show any context.

Response:

We have labeled some of the genes in Figure 1 and 3.

6. The authors infer that “.....partially explain why there was no candidate causal gene attributable to certain GWAS loci” But could it also mean that even though the SNP is associated with the trait it might be a weak effect eQTL in aggregate...for the genes in the model.

Response:

We totally agree with the reviewer on this point. We have added the following sentence in the revised manuscript.

“Notably, TWAS may also fail to identify causal genes when weak eQTLs were attributed to GWAS loci despite of association with the trait.”

7. I am not sure what evidence the authors use to make the inference that “...that the strategy of conducting TWAS in a non-trait-related tissue may identify non-causal genes even if there are hits at a locus.” Because they have not proved that those hits are non-causal in the manuscript.

Response:

Since we agree with the reviewer that TWAS analyses in other tissues are not closely related to the topic in the present study (see comment #3), we have deleted this part (i.e., text and supplementary figures) in the revised manuscript.

8. No page numbers on the manuscript. I am not sure if it is the requirement of the journal but would have made the reviewers commenting easier!

Response:

According to the reviewer’s suggestion, we have added the page number.

References

1. Bulik-Sullivan, B.K. *et al.* LD Score regression distinguishes confounding from polygenicity in genome-wide association studies. *Nat. Genet.* **47**, 291-295 (2015).
2. Bulik-Sullivan, B. *et al.* An atlas of genetic correlations across human diseases and traits. *Nat. Genet.* **47**, 1236-1241 (2015).
3. Astle, W.J. *et al.* The Allelic Landscape of Human Blood Cell Trait Variation and Links to Common Complex Disease. *Cell* **167**, 1415-1429 e1419 (2016).
4. Saha, A. & Battle, A. False positives in trans-eQTL and co-expression analyses arising from RNA-sequencing alignment errors. *F1000Res* **7**, 1860 (2018).
5. Chen, L. *et al.* Genetic Drivers of Epigenetic and Transcriptional Variation in Human Immune Cells. *Cell* **167**, 1398-1414 e1324 (2016).
6. Gusev, A. *et al.* Integrative approaches for large-scale transcriptome-wide association studies. *Nat. Genet.* **48**, 245-252 (2016).
7. Barbeira, A.N. *et al.* Exploring the phenotypic consequences of tissue specific gene expression variation inferred from GWAS summary statistics. *Nat. Commun.* **9**, 1825 (2018).
8. Gamazon, E.R. *et al.* A gene-based association method for mapping traits using reference transcriptome data. *Nat. Genet.* **47**, 1091-1098 (2015).
9. Zhu, Z. *et al.* Integration of summary data from GWAS and eQTL studies predicts complex trait gene targets. *Nat. Genet.* **48**, 481-487 (2016).

10. Pasaniuc, B. *et al.* Fast and accurate imputation of summary statistics enhances evidence of functional enrichment. *Bioinformatics* **30**, 2906-2914 (2014).
11. Urmo Võsa, A.C., *et al.* Unraveling the polygenic architecture of complex traits using blood eQTL metaanalysis. *BioRxiv* (2018).
12. Edwards, S.L., Beesley, J., French, J.D. & Dunning, A.M. Beyond GWASs: illuminating the dark road from association to function. *Am. J. Hum. Genet.* **93**, 779-797 (2013).
13. Corradin, O. *et al.* Combinatorial effects of multiple enhancer variants in linkage disequilibrium dictate levels of gene expression to confer susceptibility to common traits. *Genome Res.* **24**, 1-13 (2014).
14. Bhattacharya, A. *et al.* A framework for transcriptome-wide association studies in breast cancer in diverse study populations. *Genome Biol.* **21**, 42 (2020).

REVIEWERS' COMMENTS:

Reviewer #2 (Remarks to the Author):

The authors have carefully addressed all my major and minor comments with sufficient details and have revised the manuscript accordingly.

Reviewer #3 (Remarks to the Author):

All my concerns have been addressed